# Beyond Pixel Histories: World Models with Persistent 3D State

**Samuel Garcin** [* 1]  **Thomas Walker** [* 1]  **Steven McDonagh** [1]  **Tim Pearce** [2]  **Hakan Bilen** [1]  **Tianyu He** [2]
**Kaixin Wang** [2]  **Jiang Bian** [2]

francelico.github.io/persist.github.io

## Abstract

Interactive world models continually generate video by responding to a user's actions, enabling open-ended generation capabilities. However, existing models typically lack a 3D representation of the environment, meaning 3D consistency must be implicitly learned from data, and spatial memory is restricted to limited temporal context windows. This results in an unrealistic user experience and presents significant obstacles to downstream tasks such as training agents. To address this, we present PERSIST, a new paradigm of world model which simulates the evolution of a latent 3D scene: **environment**, **camera**, and **renderer**. This allows us to synthesise new frames with persistent spatial memory and consistent geometry. Both quantitative metrics and a qualitative user study show substantial improvements in spatial memory, 3D consistency, and long-horizon stability over existing methods, enabling coherent, evolving 3D worlds. We further demonstrate novel capabilities, including synthesising diverse 3D environments from a single image, as well as enabling fine-grained, geometry-aware control over generated experiences by supporting environment editing and specification directly in 3D space.

## 1. Introduction

Interactive world models aim to generate experiences that unfold over time in response to user actions, enabling a new class of photo-realistic, personalised, and immersive interactive experiences for human users (Kanervisto et al., 2025; Ball et al., 2025; Huang et al., 2025a). At the same time,

---
[*]Equal contribution  [1]University of Edinburgh  [2]Microsoft Research. Correspondence to: Samuel Garcin <s.garcin@ed.ac.uk>, Kaixin Wang <kaixin96.wang@gmail.com>.

*Proceedings of the 43rd International Conference on Machine Learning*, Seoul, South Korea. PMLR 306, 2026. Copyright 2026 by the author(s).

such models hold significant potential for safely training embodied agents within simulators learned directly from data (Hafner et al., 20 2; Micheli et al., 2023; Alonso et al., 2024; Garcin et al., 2024). Unlike passive video generation, interactive generation requires models to respond meaningfully to external interventions. In this setting, realism is not determined solely by per-frame visual quality, but by the degree to which the generated experience maintains persistent environments and stable dynamics throughout extended rollouts.

Most existing approaches to interactive video generation rely on autoregressive (AR) models that condition on the history of past observations and actions to generate the most recent frame, often using diffusion-based transformers with causal attention (Peebles & Xie, 2023). This formulation is attractive: it naturally supports real-time action conditioning, allows generation to proceed indefinitely, and integrates cleanly with recent advances in large-scale video modelling. As a result, AR video diffusion models form the backbone of many recent neural game engines and interactive world models (Che et al., 2024; He et al., 2025; Valevski et al., 2024b; Zhang et al., 2025; Yu et al., 2025).

However, conditioning on high-dimensional visual observations is computationally expensive, and the number of prior frames that AR models can ingest is tightly constrained by hardware. In practice, this limits AR generation to condition on what amounts to only a few seconds of past footage, even when observations are compressed to lower spatial and/or temporal resolutions using learned autoencoders. As a result, a growing body of work has explored how best to populate this limited contextual window, most commonly through heuristic strategies that retrieve individual key frames from a memory bank of past observations (Xiao et al., 2025a; Huang et al., 20 2).

Yet, an inherent flaw of key-frame retrieval methods is that each pixel observation provides only partial, redundant, and viewpoint-dependent information about the world at a fixed point in time. As the memory bank grows, identifying the relevant past evidence becomes increasingly difficult. As a result, generating extended and coherent experiences within complex 3D environments remains an open challenge.

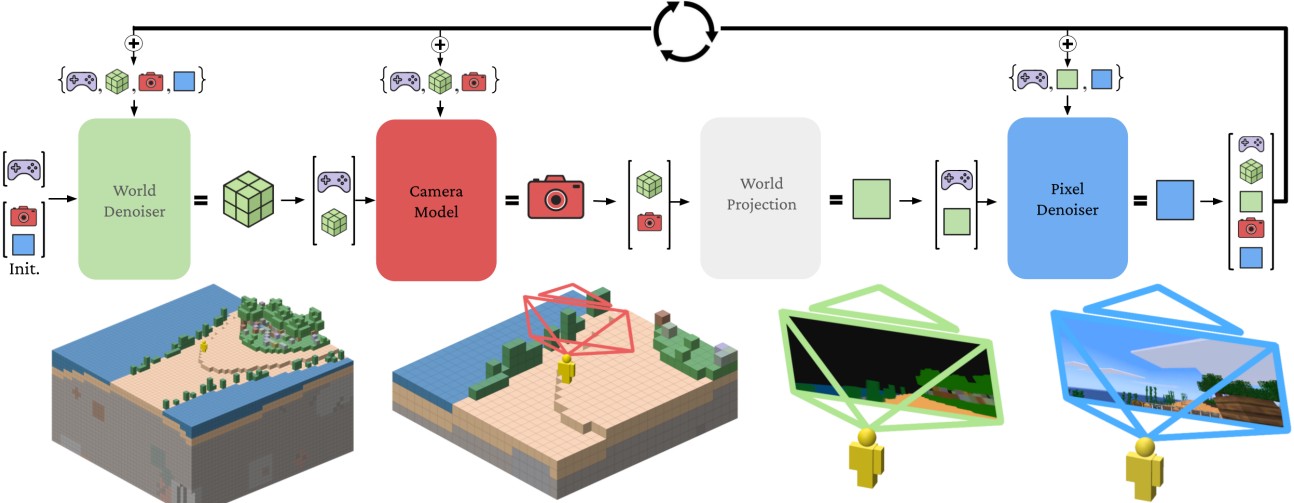

*Figure 1.* Initialized with a single pixel frame, PERSIST evolves in an autoregressive loop in response to user actions 🎮. We first denoise the 3D environment centred on the agent in the form of a latent *world-frame* 🧊. Next, camera parameters 📷 are predicted with a feed-forward transformer. We then project the world to the camera plane to form a depth-ordered stack of world latents 🟩. Finally, pixel latents 🟦 are denoised, using pixel-aligned 3D information from the world latents stack 🟩 as guidance.

In this work, we propose to depart from pixel-based histories and substitute key frame retrieval with *active key frame generation*. We are inspired by traditional 3D simulators and game engines, which maintain coherence by directly rendering pixel frames from a persistent and dynamically evolving 3D state of the world. We introduce PERSIST, a framework that brings persistence to learned AR world models by tracking a dynamic 3D representation of the environment. We describe our framework in Figure 1. PERSIST decomposes world simulation into three coupled components: a *world-frame model* that predicts how a learned representation of a 3D scene evolves over time, a *camera model* that tracks the agent's viewpoint within this scene, and a *world-to-pixel generation module* that produces observations via differentiable projection and a learned rendering function.

Rather than treating pixels as the primary carrier of memory, PERSIST models the structure and dynamics of the environment in a latent 3D space. This persistent world representation captures how the scene evolves over time, while the camera acts as a query mechanism that extracts the subset of 3D information relevant to guide the generation of the current frame. This formulation enables long-horizon rollouts with fixed-cost memory, enforces geometric consistency by construction, and remains fully learned and compatible with modern diffusion and flow-based training objectives.

We evaluate PERSIST in a complex 3D environment and find that explicitly modelling persistent 3D dynamics leads to substantially improved long-horizon generation quality compared to rolling-window and memory-retrieval baselines. Our approach exhibits stronger temporal stability, improved spatial memory when revisiting previously ob-

served regions, and markedly better 3D consistency overall. Beyond quantitative gains, persistent world representation enables new capabilities, including explicit 3D scene initialisation, mid-episode scene edits, and the emergence of off-screen dynamic processes that continue to evolve even when not directly observed.

## 2. Related Work

**World models.** World models (Ha & Schmidhuber, 2018) aim to simulate environments by predicting future states conditioned on user actions. Recent progress in video generation models enables high-fidelity interactive simulation (Ball et al., 2025; Alonso et al., 2024; Decart et al., 2024). However, maintaining spatial and temporal consistency over extended rollouts remains challenging due to the finite number of past frames existing models can incorporate as context. Prior works propose to alleviate this limitation via spatio-temporal compression of input tokens (Hafner et al., 20 2), memory-efficient attention mechanisms (Po et al., 2025) or by populating the model context window with frames sampled from the full generation history (Song et al., 2025). Among this latter class of approaches, (Xiao et al., 2025a; Huang et al., 20 2) propose sampling strategies that retrieve key-frames according to their spatial relevance. Nevertheless, such methods rely on retrieving relevant information from an ever-growing history of pixel observations, a task that becomes increasingly difficult and costly over extended rollouts. In contrast, PERSIST retrieves contextual information directly from a dynamically evolving 3D latent state, making its information retrieval mechanism independent of episode length.

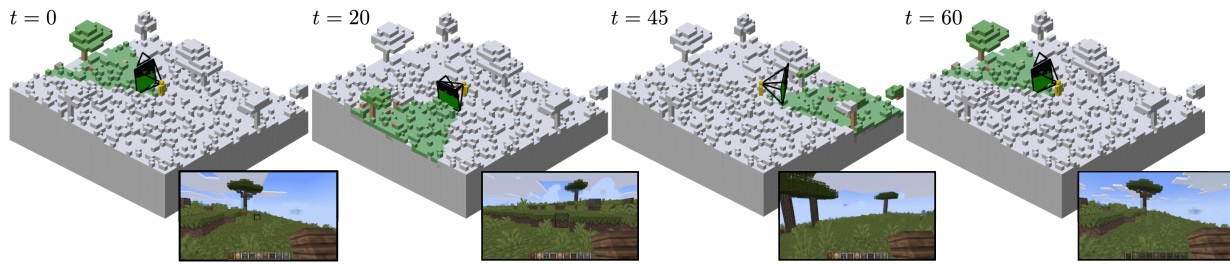

*Figure 2.* PERSIST enables long-horizon spatial memory by modelling the dynamics of a 3D world-frame 🧊 around the agent. Camera parameters 📷 then act as memory look-up key, fetching relevant features from the world frame via a geometric projection (here visualized as the coloured voxels).

**3D environment representations.** In parallel, a growing number of approaches incorporate explicit 3D representations into their generative architectures to produce explorable and spatially consistent environments (Huang et al., 2025a; Labs, 2025; Zhou et al., 2025). However, these environments remain static, restricting interaction to basic navigation within a frozen world. In contrast, PERSIST learns a 3D representation that evolves over time, enabling it to capture dynamic processes occurring within the environment, and richer agent–environment interactions.

**Neural rendering.** While neural radiance fields (NeRFs) have become a popular, compact, and resolution-free representation of 3D scenes (Mildenhall et al., 2021; Müller et al., 2022; Li et al., 2023), they rely on costly ray marching with many neural network queries, making them prohibitively expensive for interactive use. PERSIST instead takes inspiration from neural deferred shading, which rasterizes explicit geometry into screen-space features and decodes them with learned neural shaders, enabling more efficient rendering (Thies et al., 2019; Worchel et al., 2022; Walker et al., 2023; Chen et al., 2023; Deering et al., 1988).

## 3. Preliminaries

**Rectified flow matching.** Our approach is based on the paradigm of flow matching and diffusion models (Lipman et al., 2023), where the task is to restore a clean data sample $x^0$ given a noised version $x^\tau$. The noise level $\tau \in [0, 1]$ determines the degree of corruption of the data, where $\tau = 0$ denotes clean data and $\tau = 1$ indicates pure noise. Rectified flow models (Lipman et al., 2023; Albergo & Vanden-Eijnden, 2023) employ a noising process $x^\tau = (1 - \tau)x^0 + \tau x^1$ that linearly interpolates between sample $x^0$ and noise $x^1 \sim \mathcal{N}(0, \mathbb{I})$. Provided with a noised sample and corresponding noise level, the model $\mathcal{V}_\theta$ is trained to predict the velocity vector $v = x^0 - x^1$ pointing towards the clean data via the conditional flow matching objective (CFM):

$$\mathcal{L}(\theta) = \left\| \mathcal{V}_\theta(x^\tau, \tau) - (x^0 - x^1) \right\|^2 \quad \tau \sim p(\tau), \quad (1)$$

where $\tau$ is often sampled from a uniform or logit-normal distribution. During inference, random noise $x^1$ is itera-

tively denoised into a clean sample $x^0$ over $K$ sampling steps, with uniform or variable step size $d^k$:

$$x^{\tau - d^k} = x^\tau + \mathcal{V}_\theta(x^\tau, \tau)d^k, \quad \tau \leftarrow \tau - d^k. \quad (2)$$

**Autoregressive video generation.** While full-sequence video diffusion models treat videos as a single data sample to be denoised jointly at inference time, AR models treat videos as a sequence of individual frames $X_1^n = x_1, x_2, \ldots, x_n$ to be diffused sequentially.[1] AR models employ causal masking strategies during training to ensure no information travels from the future into the past. Diffusion forcing (Chen et al., 2024) improves the generation stability of AR video diffusion models at inference by introducing per-frame noise levels that are sampled independently during training. At inference, AR generation becomes a special case of the training setup, where the current frame is iteratively denoised from random noise and past frames receive a small fixed noise level $\tau_{\text{ctx}}$.

**Interactive world simulation.** Because generation proceeds step-by-step, AR models can in principle be unrolled indefinitely and conditioned on external control signals provided at each timestep. This enables a transition from passive video generation to interactive world simulation. We employ a similar formalism as Valevski et al. (2024a) to characterize the problem of learning a *world model*, which we define here as a learned simulator of some *interactive environment* $\mathcal{E}$:

$$\mathcal{E} = \langle \mathbb{S}, \mathbb{O}, \mathbb{A}, \Omega, p \rangle, \quad (3)$$

where $\mathbb{S}$ is the space of latent states, $\mathbb{O}$ the space of observations, $\mathbb{A}$ the set of actions that may be taken within the environment, $\Omega : \mathbb{S} \to \mathbb{O}$ a partial projection function mapping states to observations, and $p(s'|a, s)$ is a transition probability function such that $s, s' \in \mathbb{S} \times \mathbb{S}$ and $a \in \mathbb{A}$. Given a distribution of initial states $p_0$ and episode length $N_0$, a policy $\pi$ in charge of selecting actions in $\mathcal{E}$, and a distance metric between observation sequences $D : \mathbb{O}^n \times \mathbb{O}^n \to \mathbb{R}$,

---

[1]Notation-wise, the noise level of an individual frame $x_t^\tau$ is indicated by superscript $\tau$, its position in the sequence by subscript $t$. In the case of a truncated sequence $X_k^t$, we overload the notation to have $k$ and $t$ indicate its beginning and end.

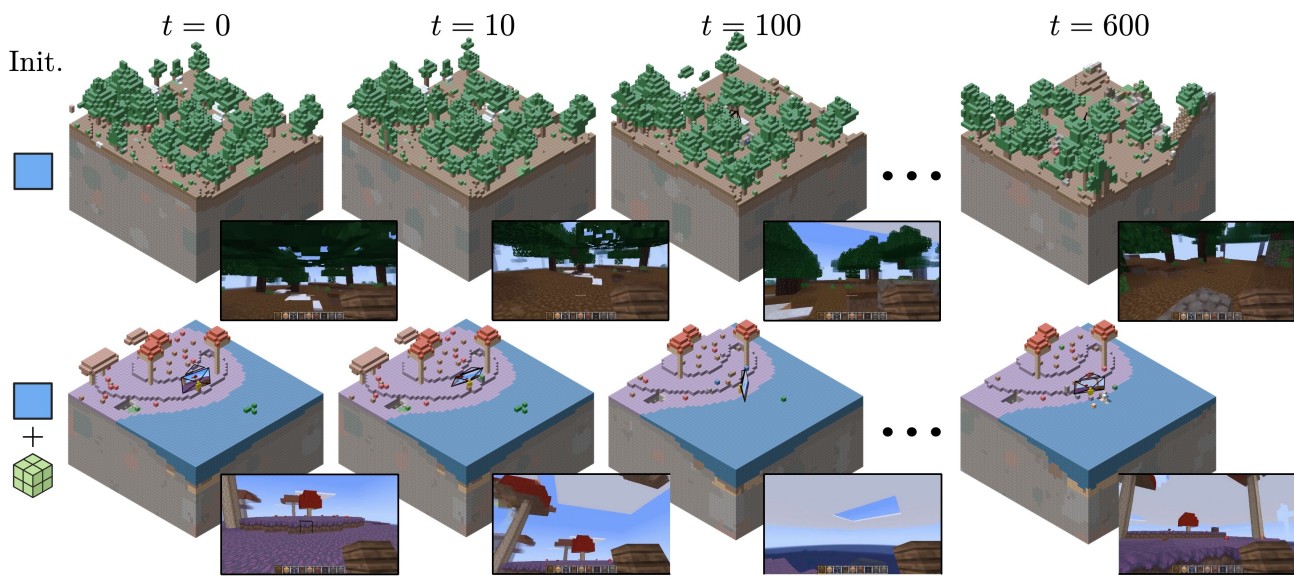

*Figure 3.* PERSIST can be initialized with a single RGB frame (■, row 1), or with a single RGB and world frame (■ + 🧊, row 2). We visualize the world-frames and videos produced by an autoregressive rollout of 600 timesteps. Even with a single RGB frame for initialization, PERSIST can generate cohesive and evolving worlds.

learning an interactive world simulation of $\mathcal{E}$ consists of minimising the objective $\mathbb{E}[D(O^n, \tilde{O}^n)]$. $O^n$ is a sequence of observations of length $n \sim N_0$ collected in $\mathcal{E}$ from starting state $s_0 \sim p_0$ and by following $\pi$; and $\tilde{O}^n$ is obtained from an AR model $\mathcal{Q}_\theta$ initialised at $\boldsymbol{o}_0 = \Omega(s_0)$ and subjected to the same action sequence.

## 4. A Persistent Environment Representation

Learning an autoregressive model $\mathcal{Q}_\theta$ is often challenging when $\mathcal{E}$ is a complex 3D environment featuring high dimensional pixel observations. First, memory constraints limit $\mathcal{Q}_\theta$ to condition on a finite context window of past $K$ frames, which degrades temporal consistency once the generated episode exceeds this horizon. Second, pixel observations typically provide partial information about the environment's hidden state, making learning accurate transition dynamics challenging. One might therefore attempt to have $\mathcal{Q}_\theta$ condition on some approximation of the hidden state $s$; however $s$ can be arbitrarily complex or entirely unmeasurable. Even if $s$ were accessible, learning the observation function $\Omega : \mathbb{S} \to \mathbb{O}$ to recover observations can be a comparably challenging problem. For example, in the case of a video game, $s$ would be the program's dynamic memory contents, which may not be particularly helpful for easily recovering pixel observations.

Instead, we define a proxy $\tilde{s} = \langle \boldsymbol{w}, \boldsymbol{c} \rangle$, where $\boldsymbol{w}$ is a *world-frame* representing a fixed region of space centred on the agent, and $\boldsymbol{c}$ is a camera state encoding the agent's view within $\boldsymbol{w}$. Although not a perfect replacement for the true hidden state, $\tilde{s}$ is an effective modelling choice when $\mathcal{E}$ is a

3D environment, as it lets us:

1. **Simplify information retrieval.** Figure 2 demonstrates how the world-frame $\boldsymbol{w}_t$ can function as a dynamic *spatial memory* module throughout the episode, while the camera state $\boldsymbol{c}_t$ acts as a *spatial lookup key* to retrieve the spatial information in $\boldsymbol{w}_t$ that is necessary to reconstruct $\boldsymbol{o}_t$. $\boldsymbol{c}_t$ encodes the information to project this information to screen-space, without requiring explicit knowledge of the true observation function.

2. **Improve dynamics predictions.** Tracking $\boldsymbol{w}$ and $\boldsymbol{c}$ over time facilitates modelling dynamics that are difficult to infer from pixel observations alone, such as interactions or collisions with out-of-view objects, or tracking how occluded regions of the environment change over time.

## 5. PERSIST

We present PERSIST (Persistent Environment Representations for Simulating Interactive Space-Time), a world simulation framework for complex 3D environments. PERSIST decomposes the world simulation objective into the tasks of *world frame prediction*, *camera prediction* and *world-to-pixel generation*.

**Dataset construction and pre-processing.** The different components of PERSIST are trained from a dataset of trajectories collected from $\mathcal{E}$. A trajectory consists of a sequence of pixel observations $O$, actions $A$, world-frames $W$ and camera views $C$. In this work, we assume that $W$ and $C$ are directly obtainable from $\mathcal{E}$, the former as a 3D voxel grid within a cuboid centred on the agent, and the latter as a vector encoding the camera intrinsics and extrinsics. Actions

are embedded as a 23-dimensional multi-hot encoding of key presses and discretised mouse movements. We train a 2D-VAE and a 3D-VAE to encode pixel- and world-frames into latent patches $\bar{o}$ and $\bar{w}$ of $10 \times 10$ pixels and $4^3$ voxels respectively.

### 5.1. World-Frame and Camera Prediction

**World frame prediction.** At each timestep, a new world-frame is generated by sampling

$$\bar{w}_t \sim \mathcal{W}_\theta(\bar{\mathbf{w}}_t | \bar{W}_{t-K}^{t-1}, A_{t-K}^t, C_{t-K-1}^{t-1}, \bar{O}_{t-K-1}^{t-1}), \quad (4)$$

where $K$ is the model's temporal context window and latents $\bar{W}_{t-K}^{t-1}$ and $\bar{O}_{t-K-1}^{t-1}$ are encoded using the VAEs. $\mathcal{W}_\theta$ is parametrised as a rectified flow model with a causal Diffusion Transformer (DiT) backbone (Peebles & Xie, 2023) employing interleaved spatial, temporal and cross-attention modules (Decart et al., 2024). We modify the spatial module to handle three spatial dimensions, and replace RoPE spatial embeddings (Su et al., 2024) with absolute position embeddings of the XYZ coordinates of the centroid of each voxel token. We keep the temporal module as-is. Actions and cameras are jointly embedded using a multi-layer perceptron (MLP). The resulting embeddings are added to the denoising timestep embedding and injected into each module via Adaptive Layer Normalisation (AdaLN) (Peebles & Xie, 2023). We concatenate pixel patches channel-wise with Plücker embeddings (Xiao et al., 2025a; Sitzmann et al., 2021) computed from $C_{t-K-1}^{t-1}$ to convey 3D projection information. The resulting patches are then injected into the model via the cross-attention module. Finally, we employ the 3D-VAE to decode $\bar{w}_t$ to its native resolution.

Crucially, $\mathcal{W}_\theta$ supports conditioning on $\bar{W} = \varnothing$, making it capable of generating the initial world frame $w_0$ from initial condition $\langle o_0, c_0 \rangle$ at inference time. Thus, while we rely on $W$ to train $\mathcal{W}_\theta$, we do not have to rely on ground truth 3D conditioning during inference. We provide examples for each inference configuration in Figure 3.

**Camera model.** At any given timestep, the cameras is represented as the 10-dimensional vector $c = \langle \mathbf{pos}, \mathbf{rot}, \text{fov} \rangle$, where $\mathbf{pos} \in \mathbb{R}^3$ encodes the camera's position in the world-frame, $\mathbf{rot} \in \mathbb{R}^6$ its orientation as a 6D continuous rotation (Zhou et al., 2019), and $\text{fov} \in \mathbb{R}$ its field-of-view. The camera model predicts $c_t = \mathcal{C}_\theta(C_{t-1-K}^{t-1}, W_{t-K}^t, A_{t-K}^t)$ using a 1D causal transformer backbone with RoPE temporal embeddings. Individual tokens are obtained by passing $\mathbf{pos}$ and $\mathbf{rot}$ through separate positional embedders, concatenating fov and applying a linear projection. $W$ is cropped to its inner-most $4^3$ voxels and embedded alongside $A$ via a joint MLP prior to AdaLN injection. Given that the camera only rotates along its pitch and yaw axis, we re-parametrise the model's output layer to predict $\bar{c} = \langle \mathbf{pos}, \Delta\text{pitch}, \Delta\text{yaw}, \Delta\text{fov} \rangle$, with residuals

$\Delta\text{pitch} = \text{pitch}_t - \text{pitch}_{t-1}, \Delta\text{yaw} = \text{yaw}_t - \text{yaw}_{t-1}$ and $\Delta\text{fov} = \text{fov}_t - \text{fov}_{t-1}$.[2] The camera model is trained via MSE losses applied to each component of $\bar{c}$.

### 5.2. World-to-Pixel Generation

In order to condition the pixel-space generation on 3D information we need to form world-to-pixel correspondence.

**World projection.** We project $w$ to screen-space via the projection operator

$$\mathcal{R}(c, w) = (\tilde{w}_{2D}, d), \quad (5)$$

where $\tilde{w}_{2D} \in \mathbb{R}^{h \times w \times l \times m}$ is a per-pixel depth-ordered list of $l$ voxel features with $m$ channels, indexed according to their position in screen space $\langle h_i, w_j \rangle$. $d \in \mathbb{R}^{h \times w \times l}$ is the linear depth of each voxel feature measured in camera space. We provide a visualisation of how $\mathcal{R}$ constructs $\tilde{w}_{2D}$ and $d$ in Figure 4. $\mathcal{R}$ employs the same $h, w$ screen dimensions as the pixel latents, ensuring pixel-level alignment with $\bar{o}$. Finally, we combine $\tilde{w}_{2D}$ and $d$ into $w_{2D} \in \mathbb{R}^{h \times w \times z}$ via channel-wise concatenation, where $z = l \times (m + 1)$.

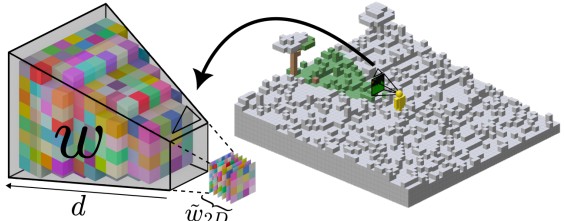

*Figure 4.* World frame $w$ features are projected to screen-space to obtain the depth-ordered stack of features $\tilde{w}_{2D}$ and linear depth information $d$.

**Pixel frame prediction.** Pixel frames are generated by sampling

$$\bar{o}_t \sim \mathcal{P}_\theta(\bar{o}_t | W_{2D}{}_{t-K}^t, A_{t-K}^t, \bar{O}_{t-K}^{t-1}), \quad (6)$$

where $W_{2D}{}_{t-K}^t$ is obtained from the projection $\mathcal{R}(C_{t-K}^t, W_{t-K}^t)$. Here, $\mathcal{P}_\theta$ acts as a learned deferred shader (Thies et al., 2019) which additionally predicts information not provided by 3D latents (e.g. texture, lighting, particle effects, screen-space overlays, ...). This choice allows $\mathcal{P}_\theta$ to learn arbitrary rendering functions, whilst encouraging spatial consistency through its dependency on $w_{2D}$. $\mathcal{P}_\theta$ is parametrised as a rectified flow model with a causal DiT backbone employing interleaved spatial and temporal modules (Decart et al., 2024). Actions are embedded using a MLP and injected via AdaLN. $w_{2D}$ is projected to latent space via a channel-wise 1D-convolution and is incorporated to $\bar{o}$ via channel-wise concatenation. Crucially, we assign more latent channels to $w_{2D}$ than to $\bar{o}$, in order to bias the model to use the 3D latent frame as its primary source of information.

---

[2] $c_t$ is recovered from $\langle \bar{c}_t, c_{t-1} \rangle$ via an algebraic manipulation at inference time.

## 5.3. Mitigating Exposure Bias

During training, the denoisers condition on latents encoded from the training data. However, they condition on autoregressive predictions at inference time, leading to exposure bias (Ning et al., 2024). We train each denoiser with diffusion forcing (Chen et al., 2024) to make them robust to the exposure bias induced by their own predictions. In addition, at inference $\mathcal{W}_\theta$ depends on latents predicted by $\mathcal{P}_\theta$, and vice-versa. To alleviate the resulting distributional shift, we apply a flat 10% random noise augmentation to $\bar{O}$ when training $\mathcal{W}_\theta$, and to $\bar{W}$ when training $\mathcal{P}_\theta$. This lets us train each component of PERSIST separately and combine them at inference time without any fine-tuning.

## 6. Experiments

We conduct our experiments within Luanti (Luanti contributors, 2024), an open-source voxel-based game engine inspired by Minecraft.[3] Voxel environments discretise 3D space into individual voxels, each voxel taking on any of thousands of possible configurations (e.g. water, stone, air, etc. ). These environments constitute a valuable research platform (Malagón et al., 2025; Fan et al., 2022; Guss et al., 2019; Baker et al., 2022), as they instantiate interpretable, complex, diverse, and persistent 3D worlds with rich player–environment interactions. Unlike prior work that learns world models from data collected within a single game map (Alonso et al., 2024; Kanervisto et al., 2025), we learn a broad distribution of procedurally generated worlds. Training world models within procedural environments substantially increases the difficulty of achieving spatial and temporal consistency, as the model cannot overfit to a fixed environment layout.

**Dataset.** We use the Craftium platform (Malagón et al., 2025) to collect game data to train and evaluate our models. We collect ∼40M environment interactions consisting of player actions, camera states, pixel and 3D observations. Actions are represented as multi-hot vectors encoding individual key presses and discretised mouse movements. 3D observations are represented as a $48^3$ voxel grid centred on the agent. In Luanti, individual voxels contain semantic information about a specific spatial location, encoded as integer labels. In total, we collect ∼100K trajectories, amounting to 460 hours of gameplay recorded at 24Hz. To interact with the game, we employ a similar strategy as (Yu et al., 2025) and design a simple policy that randomly samples from a set of pre-defined action sequences.

**Training and Inference.** All models are trained using the AdamW (Loshchilov & Hutter, 2019) optimizer with a learning rate of $1e-4$. We first train the VAEs and pre-encode

---

[3]We prefer Luanti over Minecraft due to its open-source design and expressive Lua API, which facilitates data collection and experimentation.

*Table 1.* FVD (↓) evaluated across different episode lengths (200, 400, and 600 frames). We compare our base configuration (PERSIST) against pixel-history baselines and various model ablations. PERSIST+$\boldsymbol{w}_0$ is discussed in Section 6.2.

| Method | 200 Frames | 400 Frames | 600 Frames |
|---|---|---|---|
| Oasis | 409 | 687 | 875 |
| WorldMem | 358 | – | – |
| No-3D-Upscale | 216 | 231 | 247 |
| Camera-GT | 161 | 152 | 152 |
| PERSIST-S | 159 | 170 | 179 |
| PERSIST | **129** | **141** | **148** |
| PERSIST+$\boldsymbol{w}_0$ | 80 | 93 | 104 |

*Table 2.* Mean user ratings (score range: 1-5) for videos generated from different PERSIST configurations and baselines. Bold numbers indicate the best performing methods. VF = Visual Fidelity, 3D = 3D Consistency, Temp = Temporal Consistency, Overall = Overall Score.

| Method | VF↑ | 3D↑ | Temp↑ | Overall↑ |
|---|---|---|---|---|
| Oasis | $2.1 \pm 0.1$ | $1.9 \pm 0.1$ | $1.8 \pm 0.1$ | $1.9 \pm 0.1$ |
| WorldMem | $1.7 \pm 0.09$ | $1.7 \pm 0.09$ | $1.5 \pm 0.08$ | $1.5 \pm 0.07$ |
| PERSIST-S | $\mathbf{2.8 \pm 0.1}$ | $\mathbf{2.7 \pm 0.1}$ | $\mathbf{2.5 \pm 0.1}$ | $\mathbf{2.6 \pm 0.09}$ |
| PERSIST | $\mathbf{2.8 \pm 0.09}$ | $2.5 \pm 0.09$ | $\mathbf{2.5 \pm 0.09}$ | $\mathbf{2.6 \pm 0.08}$ |
| PERSIST+$\boldsymbol{w}_0$ | $3.2 \pm 0.1$ | $2.8 \pm 0.1$ | $2.8 \pm 0.1$ | $3.0 \pm 0.1$ |

the 3D and pixel latents to accelerate the training of the dynamic models. The pixel and 3D VAEs take 4 and 12 days to train using 8 A100 GPUs. We train two versions of the 3D denoiser, "3D-S" and "3D-XL", which uses $8\times$ more spatial tokens. 3D-S takes 3 days to train on 8 H100 GPUs, while 3D-XL and the pixel denoiser each take 10 days. The camera model trains on a single A100 in 16 hours with batch size 256. All other models use batch size 64. Each model is trained separately and assembled in the full pipeline with no further fine-tuning. At inference, we use 20 denoising steps per-frame and slightly noise past context frames to make the denoisers more resilient to imperfections in their own generations. We provide more implementation details in Appendix A.

### 6.1. Interactive Generation

In this section, we evaluate our method's ability to generate spatially and temporally consistent interactive experiences within game worlds held out from the training set.

**Baselines.** We compare our approach with Oasis (Decart et al., 2024) and WorldMem (Xiao et al., 2025b), two interactive video generation methods employing the same DiT backbone as $\mathcal{P}_\theta$. WorldMem tracks camera states during the rollout and guides generation by retrieving key-frames from $O_0^{t-1}$ that closely match the current camera state. Oasis directly conditions on a sliding window of the most recent $K$ observations.

**Ablations.** We conduct several ablations to evaluate the

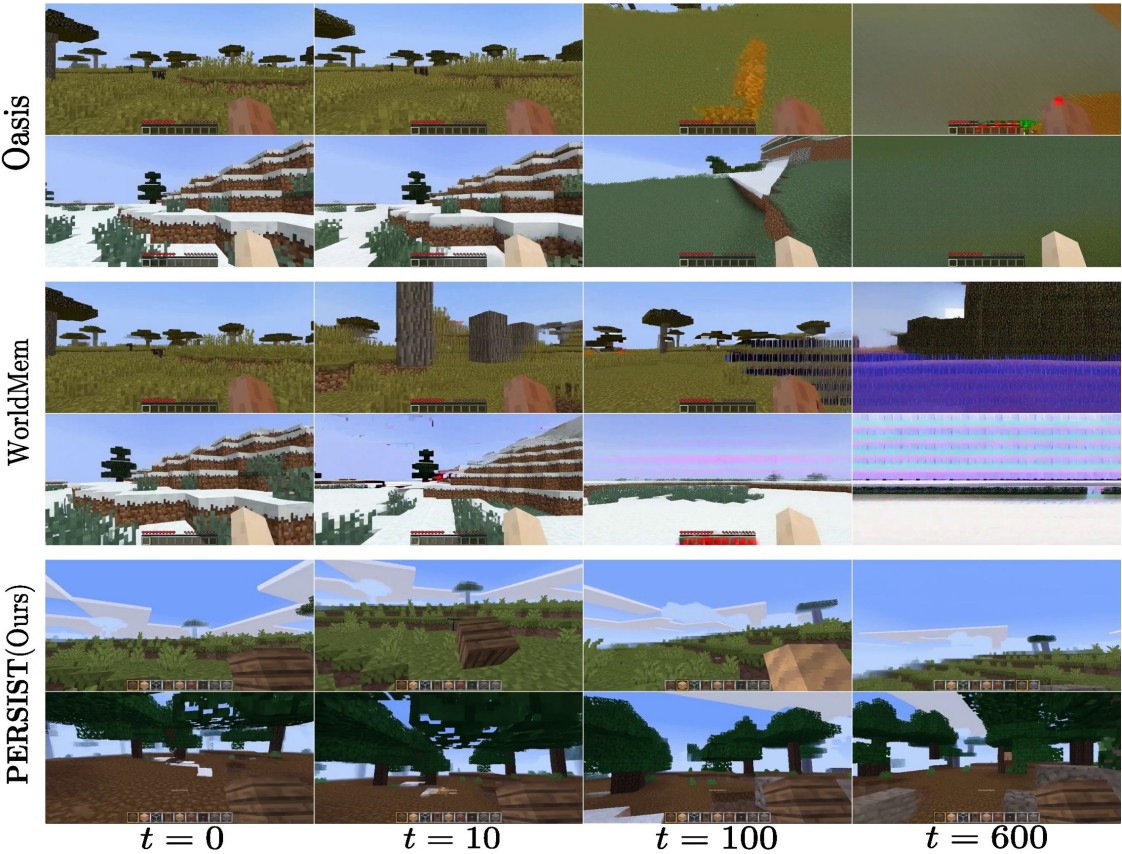

*Figure 5.* Video frames generated over 600 timestep episodes by PERSIST (Ours), Oasis (Decart et al., 2024) and WorldMem (Xiao et al., 2025a).

core components and design choices of our method. Unless otherwise specified, our base configuration and all ablations utilize the 3D-XL denoiser.

- **PERSIST-S** employs the smaller 3D-S denoiser to measure how an $8\times$ reduction in spatial tokens impacts the modeling capabilities of $\mathcal{W}_\theta$.
- **No-3D-Upscale** skips the 3D-VAE enhancement, projecting the world-frame latents directly into screen space without upscaling them to their native 3D resolution.
- **Camera-GT** bypasses our learned camera model by conditioning directly on the ground-truth cameras.

Finally, we note that Oasis effectively serves as a baseline ablation of the entire 3D conditioning mechanism.

**Evaluation Setup.** To ensure a fair comparison, all baselines and ablations utilize the same VAEs and are trained under an identical rectified flow formulation on the training set. We measure the Fréchet Video Distance (FVD) (Unterthiner et al., 2018) across 168 evaluation trajectories collected from held-out game worlds. These trajectories are generated using an action policy designed to balance world exploration with revisiting previously rendered locations from new viewpoints. Table 1 reports FVD scores computed over sets spanning the first 200, 400, and 600 trajectory timesteps. We use the first 400 ground truth frames and cameras to initialise WorldMem's memory bank. This contrasts with PERSIST, which only requires a single initial frame $\langle \boldsymbol{o}_0, \boldsymbol{c}_0 \rangle$.

**Quantitative Results.** As shown in Table 1, baselines relying solely on pixel history (Oasis and WorldMem) yield significantly worse FVD scores, with generation quality degrading sharply over time. In contrast, PERSIST and its variants maintain stable FVD scores across extended horizons, highlighting the critical role of conditioning on the 3D state. Among the ablations, No-3D-Upscale suffers the highest FVD penalty. This confirms the benefit of rendering from a high-resolution 3D latent, which produces a $\boldsymbol{w}_{2D}$ with superior depth resolution and 3D-to-pixel alignment. Conversely, PERSIST-S performs only marginally worse than the base configuration. While upscaling is crucial for rendering, the underlying 3D representation is highly robust to spatial compression (up to $512\times$ when combining the 3D-VAE and 3D-S denoiser). Finally, while Camera-GT achieves an FVD nearly identical to the base model, we observe that bypassing the learned camera model introduces physical inconsistencies that are not captured by the FVD (e.g. agent phasing through terrain or floating). We provide a qualitative example in Figure 11.

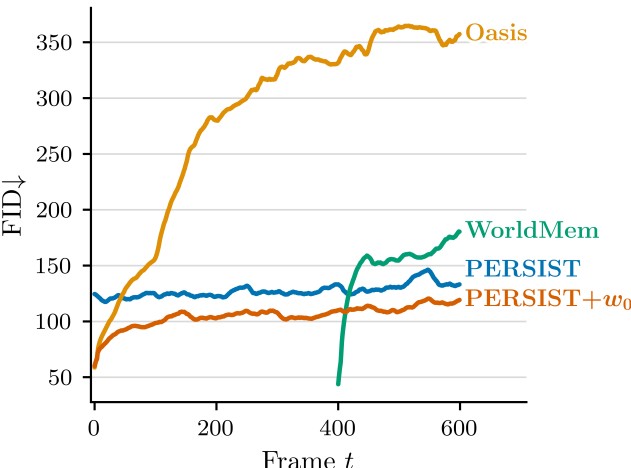

*Figure 6.* FID scores compared to ground truth over 600 frame episodes. PERSIST configurations remains stable, while baselines relying on pixel-histories degrade rapidly.

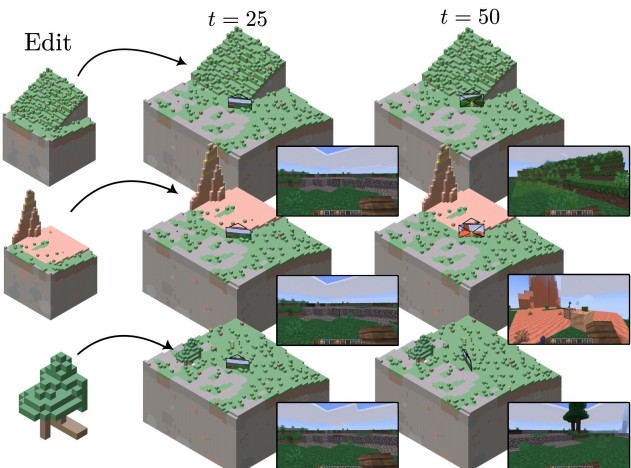

*Figure 7.* PERSIST's explicit 3D representation allows us to edit the world-state during an episode, enabling intuitive fine grained control over generations. We demonstrate this above with both global edits to terrain and biome, as well as the placement of smaller assets such as trees.

**Generation stability.** Qualitative analysis of sample rollouts (Figure 5) confirms that pixel-history baselines consistently produce irrecoverable artifacts by the 200th timestep. In contrast, PERSIST produces remarkably stable rollouts. We corroborate this observation by measuring FID (Heusel et al., 2017) at each generation timestep (Figure 6), finding that PERSIST suffers virtually no degradation over a 600-step episode. PERSIST does begin the episode with a slightly higher FID because it regenerates the initial pixel observation to ensure proper pixel/world-frame alignment. To demonstrate that this initial penalty is strictly an alignment artifact, Figure 6 includes PERSIST+$w_0$, a variant introduced in Section 6.2 that receives perfectly aligned initial world and pixel frames. Regardless of initialization, PERSIST maintains its visual quality over time, whereas pixel-history baselines rapidly degrade.

In practice, we observe that the world-frames produced by $\mathcal{W}_\theta$ remain coherent for several thousand steps. Because $w_{2D}$ is obtained directly from these highly stable world-frames, it provides reliable structural guidance over long horizons and lets $\mathcal{P}_\theta$ recover from visual artifacts. We showcase multiple examples of this self-correction over the course of a 2,000-step episode in Figure 13.

**Human study.** While FVD and FID scores capture distribution-level discrepancies, they are known to favor per-frame visual quality over the spatial and temporal coherence of individual generations (Ge et al., 2024). To complement our automated metrics, we conduct a human study comprising over 800 rollout evaluations from 28 participants. Users rated each video on "Visual Fidelity", "3D Consistency" and "Temporal Consistency", before assigning an overall score. Results are summarized in Table 2, with full methodological details in Appendix B and extended results in Appendix C. Across all metrics, PERSIST configurations consistently

outperform the baselines, corroborating our quantitative and qualitative findings. Interestingly, PERSIST-S performs similarly to the base configuration in both average user ratings and head-to-head comparisons. This highlights that the FVD penalty observed for PERSIST-S does not translate to a human-perceptible degradation in generation quality.

**Inference efficiency.** Finally, we conduct a preliminary investigation into the speed-quality tradeoff of our method. We find that generation stability is preserved even when reducing the number of denoising steps from 20 down to 2 and 4 for $\mathcal{W}_\theta$ and $\mathcal{P}_\theta$, respectively. This yields a $3\times$ inference speed-up with only a moderate impact on generation quality (see Appendix D for a detailed analysis).

### 6.2. New Applications and Emerging Capabilities

In addition to its improvement to the quality and coherence of generated experiences, we find that PERSIST's 3D representation confers a number of new capabilities.

**3D generation.** At initialisation, it is essential for $w_0$ to capture the structure and semantics of the provided conditioning RGB observation. In Figure 12, we show that $\mathcal{W}_\theta$ successfully generates plausible starting world frames, while also outpainting unseen regions differently from one episode to the next. This ability lets PERSIST generate environments that are both diverse and coherent, supporting a wide variety of interactive experiences.

**Explicit 3D initialisation.** By default, PERSIST initialises generation from starting condition $\langle o_0, c_0 \rangle$, with the initial world-frame $w_0$ being inferred by the world dynamics model $\mathcal{W}_\theta$. However, PERSIST also supports $w_0$ being

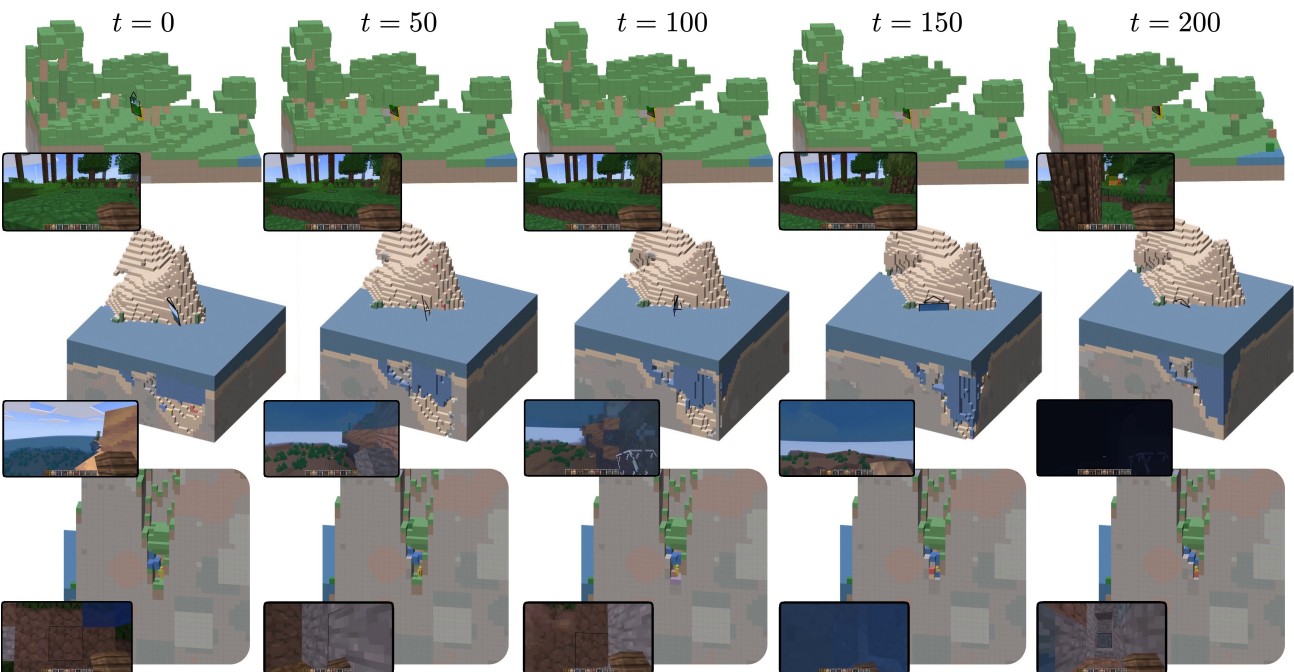

*Figure 8.* PERSIST's 3D state enables collision modelling with out-of-view elements (top: player moves backwards into a tree at $t = 50$). Since the 3D state is dynamic, it evolves even when unobserved (middle: a cave filling with water). This allows off-screen events to produce emergent on-screen effects (bottom: water flowing onto the player at $t = 150$).

provided as a starting condition[4]. This explicit 3D conditioning allows a greater degree of control over the generated experience than providing an image, as the full surroundings of the agent can be specified.

To assess how effectively PERSIST can leverage explicit 3D conditioning, we evaluate a variant denoted PERSIST+$w_0$, where a ground-truth world-frame $w_0$ is provided at initialisation. PERSIST+$w_0$ achieves lower FID and FVD scores and higher human ratings (Tables 1 and 2, Figure 6), demonstrating that the model successfully incorporates the additional information contained within $w_0$.

**World edits mid-generation.** Valevski et al. (2024a) and Kanervisto et al. (2025) proposed to perform manual edits to generated pixel frames before re-injecting them as context, as an alternative way of controlling the generated experience. As we can extract $w_t$ at any point during generation, we obtain the capability of making precise 3D-edits to the world mid-episode. We pause generation at timestep $t$ and manually edit $w_t$ to obtain $\tilde{w}_t$. We then restart generation using $\{\tilde{w}_t, c_t, o_t, a_t\}$ as the starting condition. We show several edit examples in Figure 7.

**Persistent world dynamics.** PERSIST learns to model environmental processes that evolve on their own. In Figure 8, we provide examples of environment dynamics and interactions occurring outside of the agent's view. We observe that

---
[4]See Algorithms 1 and 2 for the respective inference algorithms of PERSIST and PERSIST+$w_0$.

dynamics occurring off-screen will sometimes act as causal drivers for emergent on-screen effects.

## 7. Conclusion

In this work, we introduced PERSIST, a world modelling framework that tracks the evolution of a persistent latent 3D state. Our results demonstrate that actively generating guidance frames from this 3D state substantially improves spatial memory, temporal coherence, and overall generation quality and stability compared to conditioning on pixel-based histories. We show that tracking this 3D world state allows the model to capture environment dynamics occurring beyond the agent's view. Additionally, it lets users pre-specify, visualise, and modify the structure of generated environments, creating new ways of controlling generated experiences.

Despite these advances, PERSIST currently relies on ground-truth 3D supervision during training and maintains a finite region of 3D space in memory. Furthermore, our implementation was not optimised for inference speed and does not yet achieve real-time inference. We believe these limitations outline a clear roadmap for future research involving persistent 3D world states: in-the-wild training via synthetic 3D annotations obtained from 2D-to-3D foundation models (Wang et al., 2025; Team et al., 2025), end-to-end post-training on generated rollouts to mitigate autoregressive drift (Huang et al., 2025b), and access to unconstrained spatial memory via a 3D memory bank. We provide a detailed discussion of these directions in Appendix E.

## Acknowledgments

We thank Dave Abel, Chris Lucas, Jianfeng Xiang, Aidan Scannell, Adam Jelley and Octave Mariotti for their insightful comments and suggestions. We also thank the anonymous participants in the human study for their time.

## Impact Statement

This work advances the field of Machine Learning by improving the spatial and temporal consistency of generative world models. While our research is primarily conducted within the simulated environment of Luanti, the underlying methodologies have potential applications in video games, digital twin simulation, and embodied AI navigation, where reliable 3D scene representation is critical. We do not foresee any immediate negative societal consequences that require specific highlighting at this stage, as our model operates within a synthetic domain.

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

# A. Additional Implementation Details

## A.1. Additional Model Details

**2D VAE.** We employ a vision transformer architecture (Dosovitskiy et al., 2021), using the open-source implementation of (Decart et al., 2024). We train the VAE via an MSE reconstruction objective, alongside a KL divergence penalty coefficient of $1e - 6$.

**3D VAE.** We employ a 3D-ResNet architecture based on the implementation provided by (Xiang et al., 2025). We train the VAE via a cross-entropy loss to predict voxel class labels, alongside a KL divergence penalty coefficient of $1e - 6$.

**Differentiable Rasterizer.** For our projecting function $\mathcal{R}$, we leverage GPU-native triangle rasterization, by assigning voxel features to faces of a static voxel-grid mesh. Then, depth-peeling (Bavoil & Myers, 2008; Nagy & Klein, 2003) is applied during rasterization to produce a per-pixel, depth-ordered stack of 3D features $W_{2D}^t \in \mathbb{R}^{H \times W \times L \times C}$, where $L$ is the maximum number of layers. Since the mesh topology is fixed for a given resolution, we construct it once at initialisation, updating vertex attributes in-place to avoid redundant mesh construction.

## A.2. PERSIST and PERSIST$+w_0$ Inference Logic

We detail the inference procedures for PERSIST with and without 3D state $w_0$ initialization in Algorithms 1 and 2 respectively.

## A.3. Model and Inference Hyperparameters

We summarize the model architectures and inference hyperparameters used in our system. Table 3 and Table 4 describe the configurations of the 3D-VAE and 2D-VAE respectively, while Table 5 details the camera model architecture. We further report the world and pixel denoiser configurations in Table 7 and Table 8. Finally, Table 9 summarizes inference-time sampling hyperparameters, and Table 5 lists token statistics across models.

# B. Evaluation Methodology

## B.1. Quantitative analysis

We collect a total of 168 rollouts of 600 frames for each method. We then compute FVD and FID scores from the latent features obtained by passing rollout and ground truth clips through a pretrained Inflated 3D ConvNet (Carreira & Zisserman, 2017). We compute FID scores at each timestep, whereas we compute the FVD by splicing rollouts into 16-frame clips, discarding leftover frames.

## B.2. Human Study Data

For the human study, we collected evaluation datasets across four hand-crafted policies: *Free Play*, *Move Forward*, *Move Backward*, and *Orbit*. These policies are designed to produce trajectories that require a high level of temporal and spatial consistency to be correctly simulated by the world model.

- **Free Play:** randomly samples action from a set of pre-defined sequences, facilitating exploration.

- **Move Forward/Backward:** The agent moves forwards or backwards, periodically spinning on itself to observe its surroundings.

- **Orbit:** The agent follows a circular trajectory, while constantly adjusting its heading to look towards the centre of the circle.

For the human study, we evaluated the models released by the original authors, as we conducted it prior to training Oasis and WorldMem models on our training set. As these methods were trained using Minecraft data, we collected separate evaluation sets in Minedojo (Fan et al., 2022) to not induce domain shift, and we replicated their respective inference configurations. In the case of WorldMem, this included initialising the memory bank used to sample key-frames with 600 ground truth cameras and pixel observations. To ensure this difference in domains did not bias human ratings, we compared ratings of

*Table 3.* 3D-VAE configuration.

| Parameter | Value |
|---|---|
| input shape | [48, 48, 48, 2138] |
| number of residual blocks | 2 |
| middle channels | [32, 128, 512] |
| $\bar{w}$ shape | [12, 12, 12, 48] |
| # parameters | 138M |

*Table 4.* 2D-VAE configuration.

| Parameter | Value |
|---|---|
| input size | [360, 640, 3] |
| patch size | 10 |
| $\bar{o}$ shape | [36, 64, 16] |
| encoder dimension | 1024 |
| encoder depth | 6 |
| encoder number of heads | 6 |
| decoder dimension | 1024 |
| decoder depth | 12 |
| decoder number of heads | 16 |
| # parameters | 227M |

*Table 5.* Camera model configuration.

| Parameter | Value |
|---|---|
| rotation embedding channels | 256 |
| translation embedding channels | 256 |
| hidden size | 1024 |
| transformer blocks | 12 |
| number of heads | 12 |
| MLP ratio | 4 |
| context window size | 8 |
| # parameters | 234M |

*Table 6.* Number of spatial and temporal tokens for different models.

| Model | # Temporal Tokens | # Spatial Tokens |
|---|---|---|
| $\mathcal{W}_\theta$-S | 8 | 216 |
| $\mathcal{W}_\theta$-XL | 8 | 1728 |
| $\mathcal{P}_\theta$ | 16 | 576 |
| $\mathcal{C}_\theta$ | 8 | n/a |

*Table 7.* World denoiser hyperparameters (3D-S configuration). The XL configuration uses patch size=1 and has the same number of learned parameters.

| Parameter | Value |
|---|---|
| input shape | [12, 12, 12, 48] |
| patch size | 2 |
| hidden size | 1024 |
| cross condition hidden size | 1024 |
| depth | 12 |
| number of heads | 16 |
| MLP ratio | 4 |
| pixel condition patch size | 2 |
| context window size | 8 |
| # parameters | 686M |

*Table 8.* Pixel denoiser configuration.

| Parameter | Value |
|---|---|
| $\bar{o}$ shape | [36, 64, 16] |
| $\bar{o}$ embedder output channels | 16 |
| patch size | 2 |
| hidden size | 1024 |
| depth | 12 |
| number of heads | 16 |
| MLP ratio | 4 |
| $\bar{w}_{2D}$ shape | [36, 64, 192, 32] |
| $w_{2D}$ embedder patch size | 6 |
| $w_{2D}$ embedder stride | 4 |
| $w_{2D}$ embedder output channels | 752 |
| context window size | 16 |
| # parameters | 460M |

*Table 9.* Flow matching sampler configurations at inference time.

| Parameter | Value |
|---|---|
| minimum noise $\sigma_{\min}$ | $1e-5$ |
| denoising steps | 20 |
| denoising step scheduling function | $\tau^k = \frac{\eta k}{1+(\eta-1)k}$ |
| $\eta$ | 3 |
| noise level applied to context (past) frames $\tau_{\text{ctx}}$ | 0.02 ($\mathcal{W}_\theta$) / 0.1 ($\mathcal{P}_\theta$) |

ground-truth videos from both sets. We observed no significant preference for either domain, with a difference in average scores of less than 0.15 (3.75%).

### B.3. Human Study Protocol

The user study was deployed via a custom web interface (see Figure 9). Participants were tasked with scoring videos according to the following metrics and instructions:

- **Visual Fidelity.** Per-frame visual quality of the video. Deduct points for visual artifacts such as blurriness, color shifts, flickering, noise, or texture degradation.

- **3D Spatial Consistency.** Spatial consistency of the scene. Deduct points if objects deform, shift, or change appearance when viewed from different camera angles, or if the relative depth and spatial relationships between objects appear inconsistent as the player moves.

- **Temporal Environment Stability and Consistency.** Consistency of the environment over time. Deduct points if the scene changes unexpectedly, such as objects disappearing, reappearing, or changing structure when revisiting previously seen areas, or if the environment degrades or becomes unnaturally simplified over time.

- **Overall Quality Score.** Your subjective rating of the overall quality of the video, considering all factors above.

Ratings had the range of 1-5. Videos were presented to the user as pairs. Each pair depicted matching behaviour (free play, move forward, move backward, orbit) but different methods (baselines, PERSIST configurations, Minedojo or Craftium ground truth). Each individual user was asked to rate up to 50 videos. User assessments were subsequently aggregated to compute average scores. We consider the first three evaluation trials for each participant to count as calibration, and we discard them from the study.

Overall, a total of 28 participants evaluated over 800 videos. 93% of study participants were AI/ML practitioners and mostly active gamers (48% casual, 41% regular). 17% of active gamers had over 100 hours of playtime on Minecraft, 21% had played 10–100 hours and 28% had under 10 hours. 35% had heard of Minecraft but had never played.

## C. Extended User Study Results

### C.1. Score Differences During Head-to-Head Comparisons

Given that users rate two side-by-side videos during each trial, we can analyse how our default configuration fares in a direct head-to-head comparison against any other baseline, PERSIST configuration, or the ground truth data. We provide score comparison matrices below (one per metric) displaying the average score deltas during head-to-head comparisons.

PERSIST configurations consistently outperform the baselines in every head-to-head comparison. During head-to-head comparisons between PERSIST-S and the default configuration, PERSIST-S narrowly wins on 3D Spatial Consistency ($+0.21$), while the default configuration achieves a slightly higher Overall Quality Score ($+0.14$). The differences in Visual Fidelity and Temporal Consistency are too narrow to call for either side. While the default configuration achieves a better FVD, the results of the study indicate that humans do not observe a significant performance degradation when we decrease the number of spatial tokens allotted to $\mathcal{W}_\theta$. In contrast, PERSIST$+w_0$ wins most head-to-head comparisons against base and PERSIST-S, which indicates that providing direct 3D conditioning generally improves performance.

Finally, users consistently prefer ground-truth videos across all metrics, highlighting that even our strongest configurations remain below the quality of real trajectories. This persistent gap underscores both the difficulty of long-horizon world modelling with procedurally generated environments, and the opportunity for further advances in persistent 3D generative representations.

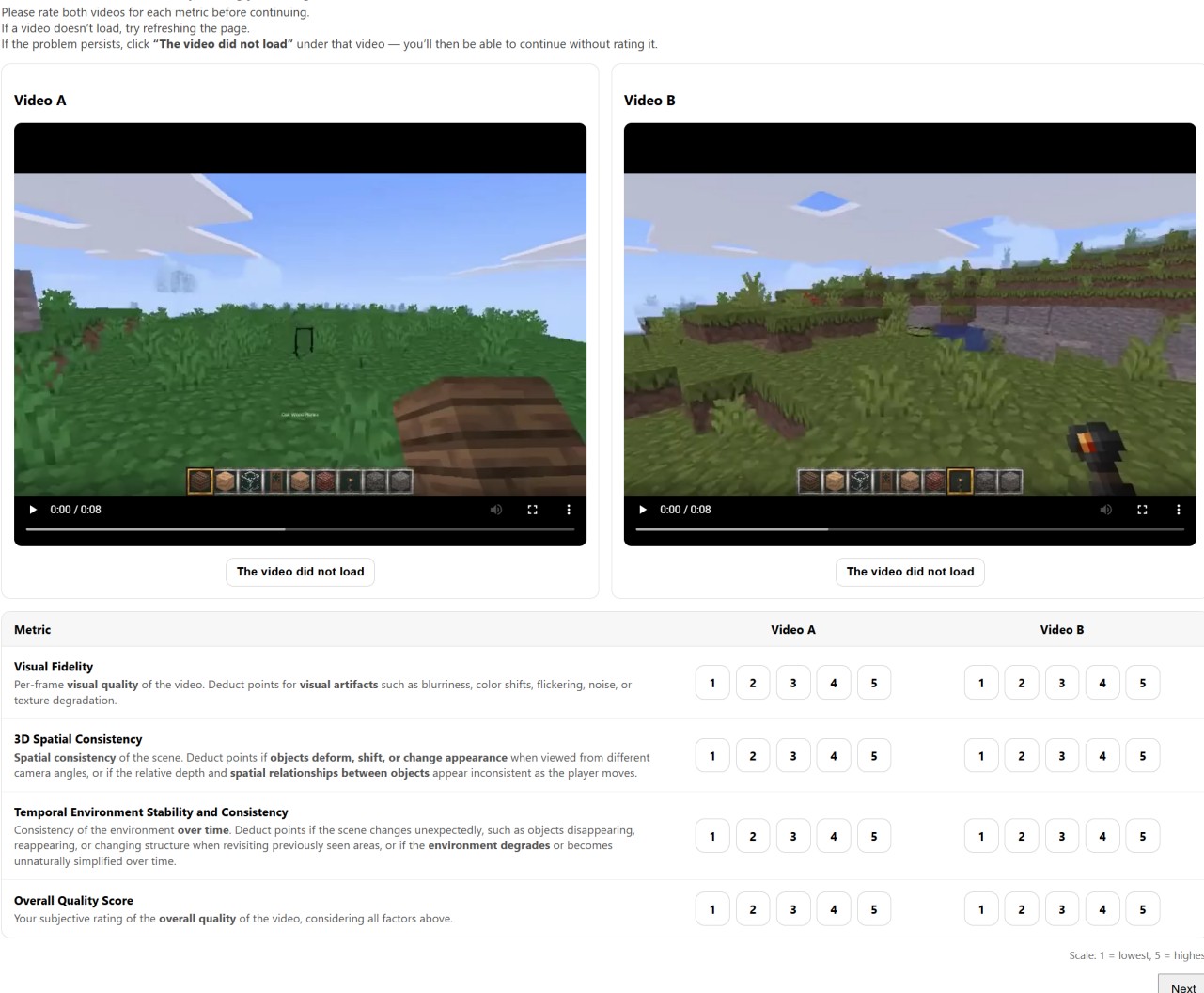

*Figure 9.* **User Study Interface.** A screenshot of our web-based platform where participants evaluate video pairs based on temporal and spatial consistency.

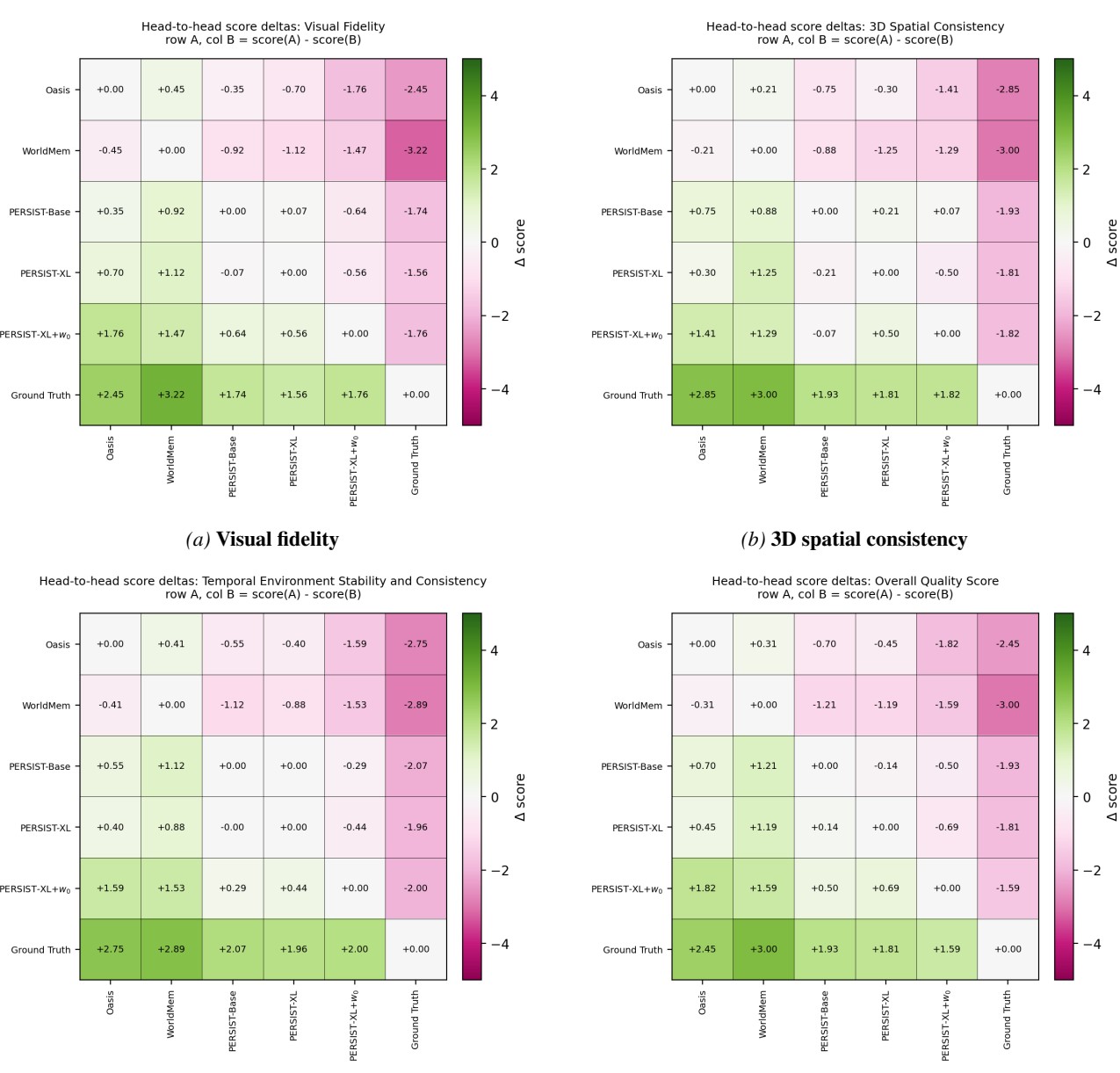

*(a)* **Visual fidelity**

*(b)* **3D spatial consistency**

*(c)* **Temporal consistency**

*(d)* **Overall quality score**

*Figure 10.* Score differences during head-to-head comparisons across evaluation metrics.

---

**Algorithm 1** PERSIST inference logic.

---

**Input:** Models $\mathcal{W}_\theta, \mathcal{C}_\theta, \mathcal{P}_\theta$, 2D-VAE, 3D-VAE,
   Model context window sizes $K_{\mathcal{W}}, K_{\mathcal{C}}, K_{\mathcal{P}}$,
   Renderer $\mathcal{R}$,
   Initial conditions $o_{\text{init}}, c_{\text{init}}$,
   Episode Length $T$,
   Agent.

1: $a_{\text{init}} \leftarrow$ NO-OP        Pipeline is always initialised using a NO-OP action.
2: $\bar{o}_0 \leftarrow$ 2D-VAE.encode$(o_{\text{init}})$
3: $\bar{w}_0 \leftarrow \mathcal{W}_\theta(a_{\text{init}}, c_{\text{init}}, \bar{o}_0)$
4: $w_0 \leftarrow$ 3D-VAE.decode$(\bar{w}_0)$
5: $w_{2D_0} \leftarrow \mathcal{R}(w_0, c_{\text{init}})$      Regenerating the starting observation from $w_0$ guarantees alignment between $w_0$ and $o_0$.
6: $\bar{o}_0 \leftarrow \mathcal{P}(w_{2D_0}, a_{\text{init}})$
7: $o_0 \leftarrow$ 2D-VAE.decode$(\bar{o}_0)$
8: Initialise rollout buffers $\{A, C, O, \bar{O}, W, \bar{W}, W_{2D}\}$ with
   $a_{\text{init}}, c_{\text{init}}, o_0, \bar{o}_0, w_0, \bar{w}_0, w_{2D_0}$
9: **for** $t$ in $1, \ldots, T$ **do**
10:    Append to $A$: $a_t \leftarrow$ Agent$(o_{t-1})$
11:    Append to $\bar{W}$: $\bar{w}_t \leftarrow \mathcal{W}_\theta(\bar{W}_{t-K_{\mathcal{W}}}^{t-1}, A_{t-K_{\mathcal{W}}}^t, C_{t-K_{\mathcal{W}}-1}^{t-1}, \bar{O}_{t-K_{\mathcal{W}}-1}^{t-1})$
12:    Append to $W$: $w_t \leftarrow$ 3D-VAE.decode$(\bar{w}_t)$
13:    Append to $C$: $c_t \leftarrow \mathcal{C}_\theta(C_{t-1-K_{\mathcal{C}}}^{t-1}, W_{t-K_{\mathcal{C}}}^t, A_{t-K_{\mathcal{C}}}^t)$
14:    Append to $W_{2D}$: $w_{2D_t} \leftarrow \mathcal{R}(w_t, c_t)$
15:    Append to $\bar{O}$: $\bar{o}_t \leftarrow \mathcal{P}_\theta(W_{2D_{t-K_{\mathcal{P}}}}^t, A_{t-K_{\mathcal{P}}}^t, \bar{O}_{t-K_{\mathcal{P}}}^{t-1})$
16:    Append to $O$: $o_t \leftarrow$ 2D-VAE.decode$(\bar{o}_t)$

---

---

**Algorithm 2** PERSIST+$w_0$ inference logic. Changes to Algorithm 1 in blue.

---

**Input:** Models $\mathcal{W}_\theta, \mathcal{C}_\theta, \mathcal{P}_\theta$, 2D-VAE, 3D-VAE,
   Model context window sizes $K_{\mathcal{W}}, K_{\mathcal{C}}, K_{\mathcal{P}}$,
   Renderer $\mathcal{R}$,
   Initial conditions $o_{\text{init}}, c_{\text{init}}, $ $w_{\text{init}}$
   Episode Length $T$,
   Agent.

1: $a_{\text{init}} \leftarrow$ NO-OP        Pipeline is always initialised using a NO-OP action.
2: $\bar{o}_0 \leftarrow$ 2D-VAE.encode$(o_{\text{init}})$
3: $\bar{w}_0 \leftarrow$ 3D-VAE.encode$(w_{\text{init}})$
4: $w_{2D_0} \leftarrow \mathcal{R}(w_{\text{init}}, c_{\text{init}})$
5: $\bar{o}_0 \leftarrow$ 2D-VAE.encode$(o_{\text{init}})$
6: Initialise rollout buffers $\{A, C, O, \bar{O}, W, \bar{W}, W_{2D}\}$ with     We can use $o_{\text{init}}$ directly as $o_{\text{init}}$ and $w_{\text{init}}$ are expected to be perfectly aligned.
   $a_{\text{init}}, c_{\text{init}}, $ $o_{\text{init}}$$, \bar{o}_0, $ $w_{\text{init}}$$, \bar{w}_0, w_{2D_0}$
7: **for** $t$ in $1, \ldots, T$ **do**
8:    Append to $A$: $a_t \leftarrow$ Agent$(o_{t-1})$
9:    Append to $\bar{W}$: $\bar{w}_t \leftarrow \mathcal{W}_\theta(\bar{W}_{t-K_{\mathcal{W}}}^{t-1}, A_{t-K_{\mathcal{W}}}^t, C_{t-K_{\mathcal{W}}-1}^{t-1}, \bar{O}_{t-K_{\mathcal{W}}-1}^{t-1})$
10:    Append to $W$: $w_t \leftarrow$ 3D-VAE.decode$(\bar{w}_t)$
11:    Append to $C$: $c_t \leftarrow \mathcal{C}_\theta(C_{t-1-K_{\mathcal{C}}}^{t-1}, W_{t-K_{\mathcal{C}}}^t, A_{t-K_{\mathcal{C}}}^t)$
12:    Append to $W_{2D}$: $w_{2D_t} \leftarrow \mathcal{R}(w_t, c_t)$
13:    Append to $\bar{O}$: $\bar{o}_t \leftarrow \mathcal{P}_\theta(W_{2D_{t-K_{\mathcal{P}}}}^t, A_{t-K_{\mathcal{P}}}^t, \bar{O}_{t-K_{\mathcal{P}}}^{t-1})$
14:    Append to $O$: $o_t \leftarrow$ 2D-VAE.decode$(\bar{o}_t)$

---

### C.2. Scores for Each Evaluation Set

*Table 10.* Per set user study scores: **circle around**.

| Method | Per Frame Visual Fidelity↑ | 3D Consistency↑ | Temporal Consistency↑ | Overall Score↑ |
|---|---|---|---|---|
| Oasis | $2 \pm 0.2$ | $1.6 \pm 0.2$ | $1.5 \pm 0.2$ | $1.6 \pm 0.2$ |
| WorldMem | $1.7 \pm 0.2$ | $1.5 \pm 0.1$ | $1.3 \pm 0.1$ | $1.4 \pm 0.1$ |
| PERSIST-S | $2.8 \pm 0.2$ | $2.6 \pm 0.2$ | $2.3 \pm 0.2$ | $2.5 \pm 0.2$ |
| PERSIST | $3.0 \pm 0.2$ | $2.5 \pm 0.1$ | $2.2 \pm 0.1$ | $2.5 \pm 0.1$ |
| PERSIST+$w_0$ | $\mathbf{3.4 \pm 0.2}$ | $\mathbf{3.0 \pm 0.2}$ | $\mathbf{2.7 \pm 0.3}$ | $\mathbf{3.1 \pm 0.2}$ |

*Table 11.* Per set user study scores: **free play**.

| Method | Per Frame Visual Fidelity↑ | 3D Consistency↑ | Temporal Consistency↑ | Overall Score↑ |
|---|---|---|---|---|
| Oasis | $2.3 \pm 0.2$ | $2.2 \pm 0.2$ | $2.0 \pm 0.2$ | $2.0 \pm 0.2$ |
| WorldMem | $1.5 \pm 0.2$ | $1.5 \pm 0.2$ | $1.2 \pm 0.1$ | $1.3 \pm 0.1$ |
| PERSIST-S | $3.0 \pm 0.2$ | $2.9 \pm 0.2$ | $2.8 \pm 0.2$ | $3.0 \pm 0.2$ |
| PERSIST | $2.9 \pm 0.2$ | $2.8 \pm 0.2$ | $2.8 \pm 0.2$ | $2.8 \pm 0.2$ |
| PERSIST+$w_0$ | $\mathbf{3.6 \pm 0.2}$ | $\mathbf{3.1 \pm 0.2}$ | $\mathbf{3.3 \pm 0.2}$ | $\mathbf{3.4 \pm 0.2}$ |

*Table 12.* Per set user study scores: **backwards and look around**.

| Method | Per Frame Visual Fidelity↑ | 3D Consistency↑ | Temporal Consistency↑ | Overall Score↑ |
|---|---|---|---|---|
| Oasis | $1.9 \pm 0.2$ | $2.0 \pm 0.2$ | $2.0 \pm 0.2$ | $1.9 \pm 0.2$ |
| WorldMem | $1.7 \pm 0.2$ | $1.8 \pm 0.2$ | $1.6 \pm 0.2$ | $1.6 \pm 0.1$ |
| PERSIST-S | $\mathbf{3.0 \pm 0.2}$ | $\mathbf{2.7 \pm 0.2}$ | $\mathbf{2.7 \pm 0.2}$ | $2.6 \pm 0.2$ |
| PERSIST | $2.7 \pm 0.2$ | $2.4 \pm 0.2$ | $2.4 \pm 0.2$ | $2.5 \pm 0.1$ |
| PERSIST+$w_0$ | $\mathbf{3.0 \pm 0.2}$ | $2.6 \pm 0.2$ | $2.6 \pm 0.2$ | $\mathbf{2.8 \pm 0.2}$ |

*Table 13.* Per set user study scores: **forward and look around**.

| Method | Per Frame Visual Fidelity↑ | 3D Consistency↑ | Temporal Consistency↑ | Overall Score↑ |
|---|---|---|---|---|
| Oasis | $2.4 \pm 0.2$ | $2.0 \pm 0.2$ | $1.8 \pm 0.2$ | $1.9 \pm 0.2$ |
| WorldMem | $2.1 \pm 0.2$ | $2.2 \pm 0.2$ | $1.9 \pm 0.2$ | $1.9 \pm 0.2$ |
| PERSIST-S | $2.2 \pm 0.2$ | $2.3 \pm 0.2$ | $2.1 \pm 0.2$ | $2.3 \pm 0.2$ |
| PERSIST | $2.6 \pm 0.2$ | $2.3 \pm 0.2$ | $\mathbf{2.3 \pm 0.2}$ | $\mathbf{2.4 \pm 0.1}$ |
| PERSIST+$w_0$ | $\mathbf{2.7 \pm 0.2}$ | $\mathbf{2.4 \pm 0.2}$ | $\mathbf{2.3 \pm 0.2}$ | $\mathbf{2.4 \pm 0.2}$ |

## D. Inference Speed Evaluation

We report inference speed comparisons of our PERSIST-S configuration against baselines in Table 14. Despite the additional overhead of tracking an explicit 3D state, PERSIST-S is faster because our implementation leverages key-value caching. We also provide a component-wise breakdown in Table 15. As expected, inference is dominated by the iterative denoising processes.

Finally, we test an inference configuration of PERSIST-S employing 2 and 4 denoising steps for $\mathcal{W}_\theta$ and $\mathcal{P}_\theta$, respectively. This results in an inference speed of 886 ms (1.13 FPS), a $3\times$ improvement over employing 20 denoising steps. Interestingly, performance degradation is limited and generation remains stable without requiring any fine-tuning or timestep distillation; FVD increases from 159/170/179 to 207/230/244 at 200/400/600 frames.

While PERSIST does not yet run in real time, we are encouraged by the minimal performance degradation of PERSIST-S and its fast inference configuration compared to the base configuration. We expect that additional speedups achieved through increased spatial and temporal compression, smaller models, and more efficient implementations could enable PERSIST to achieve real-time interactivity. We leave this investigation to future work.

*Table 14.* Inference speed comparison against baselines. All tests were conducted on a single A100 GPU with a batch size of 1 and 20 denoising steps.

| Method | Time per frame (ms)↓ |
|---|---|
| PERSIST-S | **2672.4** |
| Oasis | 3007.4 |
| WorldMem | 6387.1 |

*Table 15.* Component-wise breakdown of generation time per frame for PERSIST-S (20 denoising steps).

| Component | Time per frame (ms) |
|---|---|
| World frame | 1538.4 |
| Pixel frame | 1111.0 |
| Camera state | 21.9 |
| Projection | 0.63 |

## E. Limitations and Future Work

**Training without ground truth 3D supervision.** Our results underscore the critical role of explicit 3D representations in maintaining environmental coherence over long horizons. Currently, however, PERSIST acquires this representation via supervised training, restricting its applicability to simulators or datasets providing 3D annotations (e.g. Zhou et al., 2018; Chang et al., 2017; Yeshwanth et al., 2023). A key direction for future work is to relax this dependency by learning 3D representations directly from pixels, or by leveraging recent advances in 2D-to-3D foundation models (Wang et al., 2025; Team et al., 2025) to generate synthetic 3D annotations as a pre-processing step.

**Eliminating exposure bias.** While PERSIST yields substantial improvements in generation quality, temporal stability, and environmental coherence over baselines, human evaluators tend to prefer ground-truth videos in head-to-head comparisons, reporting a gradual degradation in visual quality and environment coherence over time. We attribute this degradation primarily to exposure bias: during training, individual pipeline components ($\mathcal{W}_\theta$, $\mathcal{C}_\theta$ and $\mathcal{P}_\theta$) are trained using ground truth data; at inference, they condition on their own past predictions and on predictions from other components, causing a train-inference distribution mismatch that grows over time. We report examples of generation artifacts that occur over the course of a 2000-step episode in Figure 13. Leveraging PERSIST's fully differentiable pipeline, future work could explore incorporating an end-to-end post-training stage on generated rollouts, thereby aligning the training and inference distributions. As demonstrated in recent work (Huang et al., 2025b), this approach can be effective in making AR models robust to their own imperfect predictions.

**Unconstrained spatial memory.** PERSIST presently tracks a finite region of 3D space centred on the agent, causing information from distant locations to be discarded as the agent moves through the environment. To retain spatial information over arbitrarily large environments, future work could consider a conditioning strategy for $\mathcal{W}_\theta$ based on loading spatial chunks from a 3D memory bank. Whereas pixel memory banks suffer from viewpoint redundancy, a 3D store is inherently deduplicated and spatially organized, offering unique advantages for efficient storage and retrieval of spatial information.

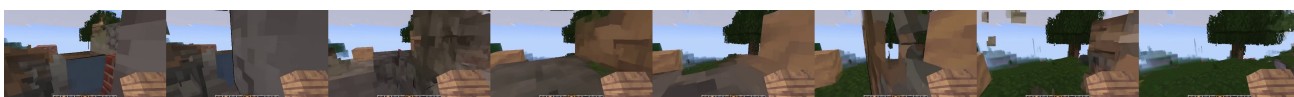

*Figure 11.* Conditioning on the ground truth camera trajectory results in physical inconsistency with the world-frame, causing the agent to phase through a wall.

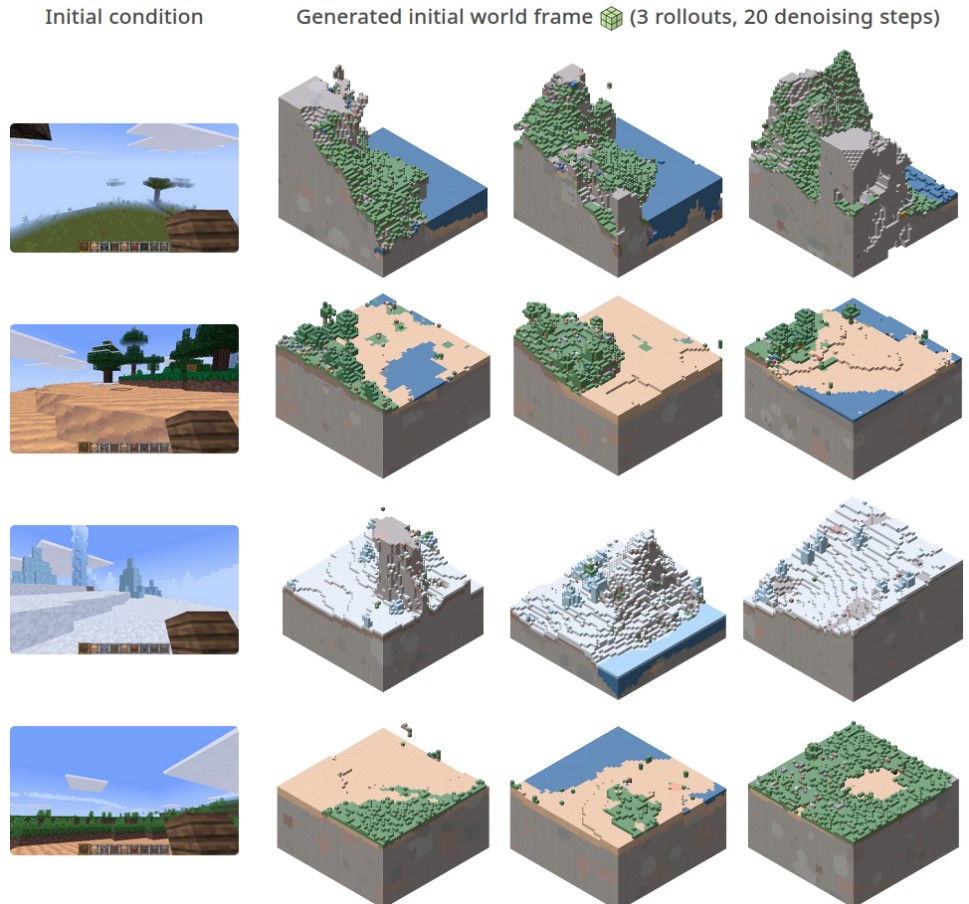

*Figure 12.* PERSIST generates diverse but coherent initial world frames from a single RGB conditioning observation. Each row corresponds to a specific input RGB frame, while each column depicts the initial world frame sampled from $\mathcal{W}_\theta$ during a specific generation episode.

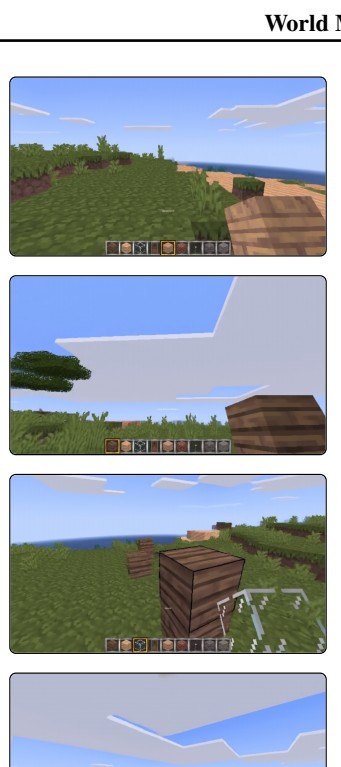

**Step 0 (0 seconds)**. Start of the episode. Generation is stable.

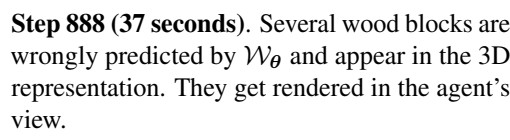

**Step 672 (28 seconds)**. A tree trunk disappears from the 3D representation. As a result only the tree foliage is rendered to the left of the agent.

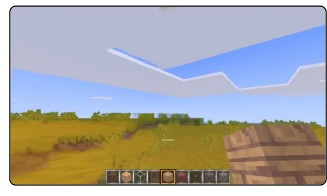

**Step 888 (37 seconds)**. Several wood blocks are wrongly predicted by $\mathcal{W}_{\boldsymbol{\theta}}$ and appear in the 3D representation. They get rendered in the agent's view.

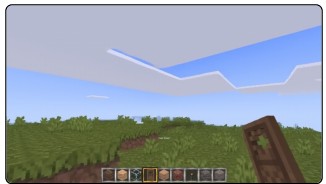

**Step 1296 (54 seconds)**. The texture of the ground is drifting to a yellowish colour. The 3D representation remains stable.

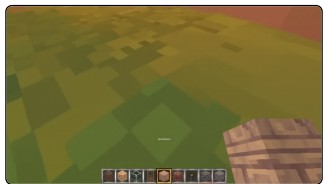

**Step 1320 (55 seconds)**. $\mathcal{P}_{\boldsymbol{\theta}}$ recovers the correct texture thanks to the grounding provided by $\boldsymbol{w}_{2D}$.

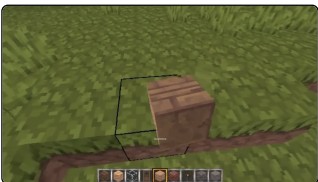

**Step 1656 (69 seconds)**. A collision gets mispredicted and the agent clips through a block. The ground texture begins to drift.

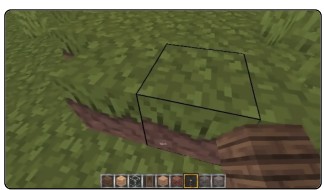

**Step 1680 (70 seconds)**. The model recovers from terrain clipping and texture drift after the agent executes a jump action.

**Step 2000 (83 seconds)**. The episode concludes on a stable frame. The environment remains globally coherent after 2000 steps, but artifacts occur more frequently due to exposure bias.

*Figure 13.* Generation artifacts and subsequent recoveries over a 2000-step (83 seconds) episode generated with PERSIST. The 3D representation drifts locally, causing individual blocks to appear and disappear from the agent's view. However we find that the 3D representation remains globally coherent and has a net stabilising effect on generation, allowing $\mathcal{P}_{\boldsymbol{\theta}}$ to recover from visual artifacts. This lets PERSIST generate rollouts lasting thousands of steps, even if glitches become more frequent over time due to autoregressive drift.

