# OpenReview forum: "Beyond Pixel Histories: World Models with Persistent 3D State"
_ICML.cc/2026/Conference — ICML 2026 regular_

### Official Review · Reviewer_qXnV · 2026-03-02

**Soundness:** 3
**Presentation:** 2
**Significance:** 2
**Originality:** 3
**Overall Recommendation:** 4
**Confidence:** 3

**Summary:**

This work addresses an important issue that in long horizon video generation, models always can't keep the 3D consistency because they just learn 3D infor implictly. This work imports a latent 3D scene which can achieve substantial improvements in spatial memory, 3D consistency. This work includes 3 components, first one is World-Frame Model which contain a 3D voxel feature representation and use a denoisy mode to predict the evolutionary process. The second part is a camera model which use a transformer to predict camera params in each step. The third part is a World-to-Pixel Generation which can differentiablly render World-Frame to 2D feature map and use a model to generate the final image based on feature map.

**Compliance With Llm Reviewing Policy:**

Affirmed.

**Final Justification:**

PERSIST proposes a genuinely novel paradigm shift from pixel-based history to persistent latent 3D state for interactive world models. The architecture is well-motivated, the experimental results within Luanti are strong (FVD 148 vs. 875 for Oasis), and inference speed is competitive. The rebuttal resolved W3 (inference speed) and honestly acknowledged W2 (fixed spatial extent). However, W1 (evaluation limited to a single voxel-based environment) remains my primary concern, as there is no evidence the approach generalizes to continuous geometry or real-world scenes. I raise my score to 4, as the idea and in-domain results merit recognition, but I flag generalization as a significant open question that the paper should discuss more prominently.

**Key Questions For Authors:**

1. Could you please extend your work to other datasets?
2. Could you please explain the limitation about Fixed spatial extent of the world-frame?
3. Could you please report the speed of inference?

**Limitations:**

Yes

**Strengths And Weaknesses:**

Strengths:
1. The motivation is well-defined and the structure is novel. This work imports an implicit 3D status to maintain the long horizon 3D consistency.
2. The experiment results are very stong to show the performance.
3. The paper is well-writen and easy to follow.
4. Several design choices reflect careful thinking: using differentiable rasterization to bridge 3D and 2D spaces while maintaining geometric consistency, the exposure bias mitigation strategy via 10% noise augmentation that allows independent module training without joint fine-tuning

Weaknesses:
1. The scope of evaluation is too limited. All of experiments are done in Luanti (just like Minecraft).
2. Fixed spatial extent of the world-frame is a fundamental limitation that is insufficiently discussed.
3. This work doesn't report the inference speed.
4. PERSIST requires ground-truth 3D world-frames and camera poses during training, which limits its applicability to settings with access to game engines or simulators

---

> ### Author Rebuttal · Authors · 2026-03-31
>
> We thank the reviewer for their detailed evaluation and for highlighting key limitations of our approach. Below we address the concerns. (References are omitted below due to space limit.)
>
> **Reply to Weakness 1**: We agree that evaluation on a single domain is an important limitation, and we now make this more explicit in the revised manuscript. To better contextualize this design choice, we also added a dedicated discussion explaining why Luanti is a particularly suitable testbed for our approach.
>  * **Why Luanti:** This domain tests if PERSIST can model organically evolving environments. Crucially, it allows us to evaluate the model’s capture of off-screen dynamics that eventually impact the player, like cascading waterfalls or flooding caves, as demonstrated here (https://figshare.com/s/f51892b14195f8bba13a).
>  * **Limited alternatives:** Many commonly used 3D video datasets provide camera motion and 3D annotations, but contain static environments, making them less informative for evaluating a model tracking the evolution of a 3D state over time. Among standard candidates, we found only limited support for domains jointly incorporating 3D annotations, a camera state, and dynamic environment evolution, as summarized below.
>
> | Domain | 3D annotations | Camera State | Dynamic environment |
> |:--|:--:|:--:|:--:|
> | RealEstate10K | ✗ | ✓ | ✗ |
> | ScanNet | ✓ | ✓ | ✗ |
> | ARKitScenes | ✓ | ✓ | ✗ |
> | NuScenes | ✓ | ✓ | ✓ |
> | Luanti (ours) | ✓ | ✓ | ✓ |
>  * **Future Work:**  We agree that extending the framework to additional domains is an important next step. In particular, NuScenes appears to be a promising candidate because it includes 3D annotations, camera state, and dynamic scene elements. However, transitioning to a real-world driving dataset requires substantial domain-specific engineering (such as managing multiple different sensor modalities), which we cannot feasibly complete within the short rebuttal window. We designed our Luanti experiments specifically to isolate our core contribution: modeling a complex and evolving interactive 3D environment while maintaining spatial and temporal consistency. We leave the adaptation to real-world datasets like NuScenes as an immediate next step for future work.
>
> **Reply to Weakness 2**: We fully agree. In fact, we recognized this fundamental limitation in the original manuscript (Line 410, right), noting that tracking a fixed cuboid means spatial information is discarded as the agent moves. To address this, we outlined a concrete path forward (Line 413): leveraging the explicit 3D structure to dynamically load and unload spatial chunks from a memory bank. To make this limitation impossible to miss, we have now dedicated a distinct paragraph to it in the revised manuscript.
>
> **Reply to Weakness 3**: We thank the reviewer for pointing this out. Below is a comparison against baselines (A100 GPU, batch size 1, 20 denosing steps):
> | (20 denoising steps) | PERSIST | Oasis | WorldMem |
> |:--|:--:|:--:|:--:|
> | Time per frame (ms) | 2672.4 | 3007.4 | 6387.1 |
>
> Despite the additional overhead of tracking an explicit 3D state, PERSIST is faster than these baselines because its architecture leverages modern speed-ups like KV-caching. We also provide a component-wise breakdown to isolate costs:
> | (20 denoising steps) | World frame | Camera state | Projection | Pixel frame |
> |:--|:--:|:--:|:--:|:--:|
> | Generation time per frame (ms) | 1538.4 | 21.9 | 0.63 | 1111.0 |
>
> **Reply to Weakness 4**: We completely agree that the current reliance on dense 3D supervision limits the applicability of PERSIST to certain domains, and we now make this limitation more explicit in the revised manuscript. Our intent in this paper is not to claim a universally applicable video world model framework, but rather to study whether maintaining an explicit 3D state improves spatial memory and long-horizon consistency **when 3D annotations are available during training**. We clarify this limitation and its possible extensions in the manuscript (Line 400, right):
> > PERSIST currently assumes access to ground-truth 3D information and camera poses during training, which restricts its applicability to settings where such information can be obtained, for instance from game engines, simulators, or datasets containing 3D annotations (Zhou et al., 2018; Chang et al., 2017; Yeshwanth et al., 2023). An important direction for future work is to relax this requirement by leveraging recent advances in 2D-to-3D foundation models (Wang et al., 2025; Team et al., 2025) to estimate world structure and camera parameters directly from pixel observations as a pre-processing step.
>
> More broadly, we view this paper as evidence that maintaining an explicit 3D representation is a useful inductive bias for world models, and thus that extending such approaches beyond settings with direct 3D supervision should be an important direction for future research.

---

> > ### Author Rebuttal · Reviewer_qXnV · 2026-04-03
> >
> > I thank the authors for the rebuttal. W3 (inference speed) is fully resolved, and I appreciate the honest acknowledgement of the fixed spatial extent limitation (W2).
> >
> > However, W1 remains my primary concern. The explanation of why Luanti was chosen is reasonable, but the core question is whether this approach generalizes beyond voxel-based environments to continuous geometry and complex lighting. The rebuttal mentions NuScenes as future work but provides no evidence. Given that the method also requires GT 3D world frames (W4), the gap from Luanti to real-world scenes could be substantial.
> >
> > Could the authors provide any preliminary evidence, even qualitative, on a non-voxel environment? This would significantly strengthen the paper's contribution.
> >
> > I maintain my score of 3.

---

> > > ### Author Response · Authors · 2026-04-05
> > >
> > > We thank the reviewer for their continued engagement and for acknowledging our clarifications.
> > >
> > > **W1: Non-voxel Environments**
> > >
> > > Due to time constraints, we are unable to demonstrate PERSIST on an entirely new dataset. However, we would like to emphasize that Luanti, while stylistically themed around voxels, incorporates a substantial number of non-voxel elements with continuous geometries. These include, but are not limited to: shrubs, flowers, reeds, doors, flowing water and fire.
> > >
> > > In addition, Luanti supports materials with varying opacities (e.g., glass and water), as well as fog effects and diffuse shadows, providing a challenging range of rendering effects.
> > >
> > > We provide some qualitative examples of PERSIST successfully modeling these components at the following link:
> > >
> > > * https://figshare.com/s/d9eaf1be35c1948a2c6e
> > >
> > > We hope this evidence helps address the reviewer’s concerns regarding non-voxel geometry, and we respectfully ask them to reconsider their assessment in light of this.

---

### Official Review · Reviewer_yWk7 · 2026-03-10

**Soundness:** 4
**Presentation:** 4
**Significance:** 3
**Originality:** 3
**Overall Recommendation:** 5
**Confidence:** 5

**Summary:**

This work introduces a pipeline that can render camera and action conditioned video model (world model) with 3D consistency, enabled through a 3D world state being updated during generation. The ability to update world state explicitly is impressive.

**Compliance With Llm Reviewing Policy:**

Affirmed.

**Key Questions For Authors:**

stated above.

**Limitations:**

1. 3D voxel as world representation could be too dense and bounded, making it impossible to do infinite-length world state tracking
2. only conduct exps on toy Minecraft dataset.
3. training still requires GT camera pose annotation, which can be inaccurate. There's some works like E-Rayzer being able to have some latent camera representation, which could be future direction.

**Strengths And Weaknesses:**

Strengths stated above. Weakness: I don't think this paper have significant flaws, while I have a few minor questions:

1. conceptually, why does the camera model need to be conditioned on the world state? did you ablate this?
2. also, why does the pixel denoiser need to be conditioned on the action signal?

Please justify the above design choices, which seems kinda arbitrary from my side.

---

> ### Author Rebuttal · Authors · 2026-03-31
>
> We sincerely thank the reviewer for their positive assessment of our work and for taking the time to provide thoughtful feedback. We address your clarifying questions regarding our conditioning mechanisms below.
>
> **Reply to Question 1**: Predicting the next camera state requires modeling the agent's environment-dependent kinematics. Action signals alone are insufficient because the agent's movement is constrained by its physical surroundings. For instance, the agent might collide with a solid object or fall into a hole. Conditioning on the explicit 3D world state provides the spatial context necessary to resolve these physical interactions, even for geometry that is currently out-of-view.
>
> Yes, we explicitly ablated this design choice. We evaluated configurations where the camera model received the world state alone, the pixel observation alone, both, or neither. The outcome of this investigation confirmed that conditioning on the 3D world state is both necessary and sufficient to accurately model the full range of movement dynamics. We will clarify this in the manuscript.
>
> **Reply to Question 2**: This conditioning is required because certain actions immediately alter the agent's visual observation without modifying the underlying 3D world state. A primary example is taking the action to switch the currently held item: this changes what is rendered in the agent's hand on-screen, but this purely visual update is not captured by the global voxel state. Action conditioning ensures the pixel denoiser can accurately render these instantaneous visual changes.

---

> > ### Author Rebuttal · Reviewer_yWk7 · 2026-04-02
> >
> > Thank you for your response. My concerns have been fully resolved, and I will maintain my current positive score.

---

### Official Review · Reviewer_kaLf · 2026-03-13

**Soundness:** 2
**Presentation:** 2
**Significance:** 3
**Originality:** 3
**Overall Recommendation:** 4
**Confidence:** 3

**Summary:**

The paper proposes PERSIST, a world model that keeps an explicit persistent latent 3D state instead of relying purely on a short window of past pixels. The system has three parts: a world-frame model for the local 3D scene, a camera predictor, and a world-to-pixel module that projects the 3D state and renders the next frame. The experiments are in voxel environments built with Luanti/Craftium, and the paper reports clear gains over Oasis and WorldMem, plus an interesting explicit 3D initialization result.

**Compliance With Llm Reviewing Policy:**

Affirmed.

**Final Justification:**

I admire the authors' efforts for the detailed rebuttal, and my main concerns are addressed by the last reply comment. I decide to raise my rating to positive.

**Key Questions For Authors:**

- Can the authors provide cleaner ablations for the roles of persistent 3D state tracking, projection, camera modeling, exposure-bias mitigation, and model scale?

- How many raters and pairwise judgments were used in the user study, and were the reported differences statistically significant?

**Limitations:**

yes

**Strengths And Weaknesses:**

**Strengths**

- The main idea is good. Replacing keyframe retrieval from pixel history with an explicit persistent 3D latent state is a meaningful shift, not just a small architectural tweak.

- The architecture is coherent. The decomposition into world-frame prediction, camera prediction, and world-to-pixel rendering is well motivated, and the differentiable projection step is a particularly elegant way to connect 3D structure with pixel generation.

- The paper does more than show FVD. The human study on visual fidelity, 3D consistency, and temporal consistency is useful here.

**Weaknesses**

- My main issue is with the comparison story. The paper says Oasis and WorldMem are evaluated on a separate MineDojo set, while PERSIST is presented in the Craftium/Luanti setting. That makes Table 1 harder to read as a clean apples-to-apples comparison.

- The method uses fairly strong privileged supervision at training time: ground-truth 3D world frames and camera poses. That is fine for a simulator paper, but it also narrows the scope quite a bit.

- The necessity of a separate learned camera model is not evaluated in this setting. In a voxel first-person environment, a deterministic or rule-based camera rollout seems like a plausible alternative, but the paper does not compare against that option.

- Efficiency is under-discussed. The paper reports substantial training cost, but does not really tell me what the inference-time tradeoff looks like compared to the baselines.

---

> ### Author Rebuttal · Authors · 2026-03-31
>
> We sincerely thank the reviewer for their constructive feedback and for pushing us to strengthen our empirical evaluation. Below we address the concerns.
>
> **Reply to Weakness 1**: We agree on the importance of a same-domain comparison and evaluated baselines trained directly on our Luanti dataset. As Oasis did not release training code, we replicated their model within our setup. Qualitatively, results show poor spatial consistency and instability after ~200 timesteps. The FVD (875) remains significantly worse than all three PERSIST configs. We are currently training the WorldMem baseline and will update results once finished.
>
> **Reply to Weakness 2**: We agree that reliance on dense 3D supervision is a primary bottleneck. As acknowledged (Line 400, right column), we view relaxing this requirement as a critical next step and will restructure the limitation and future work sections to clarify this.
>
> **Reply to Weakness 3**: A learned camera model is essential because mapping actions to movement involves state-dependent dynamics, such as varying physics (move on land vs. water) and intricate collisions, that are difficult to hand-engineer. Conditioning directly on ground-truth (GT) camera states fails because PERSIST's 3D world is initialized from a single observation (with unseen or occluded areas) and thus does not perfectly align with the GT environment. Forcing a GT trajectory causes the agent to clip through geometry (example rollout https://figshare.com/s/9ba7beb6be8e3829efde), disrupting generation and accumulating artifacts. In contrast, our learned model ensures the trajectory remains physically consistent with the generated geometry.
>
> **Reply to Weakness 4**: We thank the reviewer for pointing this out. Below is a comparison against baselines (A100 GPU, batch size 1, 20 denoising steps):
> | (20 denoising steps) | PERSIST | Oasis | WorldMem |
> |:--|:--:|:--:|:--:|
> | Time per frame (ms) | 2672.4 | 3007.4 | 6387.1 |
>
> Despite the additional overhead of tracking an explicit 3D state, PERSIST is faster than these baselines because its architecture leverages modern speed-ups like KV-caching. We also provide a component-wise breakdown to isolate costs:
> | (20 denoising steps) | World frame | Camera state | Projection | Pixel frame |
> |:--|:--:|:--:|:--:|:--:|
> | Generation time per frame (ms) | 1538.4 | 21.9 | 0.63 | 1111.0 |
>
> **Reply to Question 1**: We have conducted ablations to validate each component of PERSIST. Given the time constraints of the discussion period, we report the FVD for these ablations on a single evaluation set (Free-play), and indicate when an experiment is still in progress. The final manuscript will include a comprehensive evaluation across all sets.
> * **a) Pixel DiT:** Since the screen-space denoiser $\mathcal{P}_\theta$ employs the same architecture as the Oasis DiT, the baseline discussed in above "reply to Weakness 1" also acts as an ablation (no 3D state tracking, no projection). FVD significantly **degrades from 148 (PERSIST-XL) to 875**.
> * **b) No voxel upscaling:** We do not upscale the world-frame to its native resolution (using the W-VAE decoder) when projecting it to screen-space. **FVD 148 -> 247**.
> * **c) No camera model:** Replaces the learned camera model with ground-truth camera rollout data. As discussed in R3, replacing the learned camera model with ground-truth rollouts causes severe physical inconsistencies (e.g., clipping through geometry). While standard distribution-level metrics like FVD are largely blind to these specific geometric clipping artifacts (**FVD 148 vs 152**), the qualitative degradation can be severe, as demonstrated in the video linked above.
> * **d) No exposure bias mitigation:** We remove the random noise augmentation applied during training (see section 5.3). It is possible that training will not have completed by the end of the discussion period, as this ablation requires retraining both the pixel and world frame denoisers from scratch.
> * **e) Model scale:** we cover this aspect in the original manuscript by comparing the PERSIST-S and PERSIST-XL configurations.
>
> **Reply to Question 2**: The user study included 28 participants completed >300 pairwise evaluations (600 ratings). A Welch's t-test confirms that all PERSIST configurations significantly outperform the baselines across all metric categories $(p < 10^{-5})$. We will add these demographic details to the appendix:
> > Participants were 93% AI/ML practitioners and mostly active gamers (48% casual, 41% regular). Regarding Minecraft/voxel game familiarity: 35% were familiar but had not played, 28% had <10 hours, 21% had 10–100 hours, and 17% had >100 hours of playtime.

---

> > ### Author Rebuttal · Reviewer_kaLf · 2026-04-03
> >
> > I thank the authors for their detailed rebuttal. The justification for the learned camera model is sound and fully resolves my concern regarding its necessity to prevent geometric clipping. However, critical empirical issues remain:
> >
> > 1. Baseline Comparability: The self-replicated Oasis model (FVD 875) appears severely degraded, acting as an unconvincing "strawman" baseline. Coupled with the incomplete WorldMem results, the core comparative claims on the target dataset remain unsupported.
> >
> > 2. Interactive Efficiency: Despite being relatively faster than the baselines, a latency of ~2.6 seconds per frame fundamentally contradicts the real-time requirements of an "interactive world simulator."
> >
> > 3. Insufficient Ablation Metrics: The ablations lack the exposure bias experiment and rely exclusively on FVD. As the authors themselves explicitly noted, FVD is "blind to geometric clipping artifacts." Consequently, reporting only FVD fails to demonstrate how each component specifically contributes to the spatial and temporal consistency claimed in the original submission. Full evaluation metrics are strictly required for proper attribution.
> >
> > Because the primary baselines are either degraded or incomplete, and the ablation metrics do not align with the core claims of 3D consistency, the empirical foundation remains insufficient. I am maintaining my score of 3 (Weak Reject) and slightly lowering my confidence score to reflect the uncertainty introduced by the self-replicated baselines.

---

> > > ### Author Response · Authors · 2026-04-07
> > >
> > > We thank the reviewer for their continued engagement and for acknowledging our clarification fully resolved their concern on the importance of the camera model. We address remaining empirical concerns below.
> > >
> > > **1. Baseline Comparisons**
> > >
> > > **Oasis**: To clarify our discussion, we refer to our Luanti-trained Oasis-DiT as *PixelDiT*, and the MineDojo-trained Oasis-DiT as *Oasis*.
> > >
> > > Qualitatively, both Oasis and PixelDiT produce generations of comparable initial quality, but both suffer from severe instability over long horizons. We attribute the moderate FVD gap between Oasis (MineDojo) and PixelDiT (Luanti) to domain differences (706 vs 875). Furthermore, we observe that Oasis often degrades into a blurry screen, while PixelDiT diverges to low-light observations. We hypothesize the latter may penalize the FVD score more heavily.
> > >
> > > * **Oasis (MineDojo) sample episode:** https://figshare.com/s/cee47e3f050eeebe2d80
> > > * **PixelDiT (Luanti) sample episode:** https://figshare.com/s/6f008b40400c524a9269
> > >
> > > To demonstrate that PixelDiT remains a robust baseline, we evaluated FVD across different episode lengths (200, 400, and 600 frames) to isolate generation quality *before* the onset of instability:
> > >
> > > | Model | 200 frames | 400 frames | 600 frames |
> > > | :--- | :---: | :---: | :---: |
> > > | **PERSIST** | **129** | **141** | **148** |
> > > | WorldMem | 358 | - | - |
> > > | PixelDiT (Pixel-only ablation) | 409 | 687 | 875 |
> > > | No voxel upscaling ablation | 216 | 231 | 247 |
> > > | Ground-truth camera ablation | 161 | 152 | 152 |
> > >
> > > At 200 frames, PixelDiT rollouts are stable, yet PERSIST still maintains a significantly lower FVD (129 vs. 409). While PixelDiT's FVD sharply increases over time, PERSIST and its 3D-aware ablations remain stable.
> > >
> > > **WorldMem** To complete the baseline comparisons, we additionally train and evaluate WorldMem on the Luanti dataset. Because WorldMem requires a substantial number of context initialization frames, we use 400 of the 600 evaluation frames as context. Following WorldMem, we evaluate using ground-truth camera trajectories.
> > >
> > > Qualitatively, WorldMem demonstrates improvements over the PixelDiT baseline, but we observe similar divergent behaviour across both datasets:
> > > * https://figshare.com/s/ac1ad8c8d99853236a2d
> > >
> > > Quantitatively, we still observe notably improved performance of PERSIST over Worldmem (FVD 129 vs 358) even for 200 frame generations where less drifting has occured.
> > >
> > > In fact, we observe that PERSISTS's 3D guidance actively pulls the model back from visual drifting. In this 2000-frame episode, the ground texture begins to drift yellow but successfully recovers around frame 1300 (54s): https://figshare.com/s/d700aef07cb7b34487de. We have added this temporal FVD analysis to the manuscript. We also use this 2000 frame rollout to directly showcase some of the artifacts described in the limitations (l426, left).
> > >
> > > **2. Interactive Efficiency**
> > > We will update the manuscript to clarify our terminology: by "interactive," we mean the generation is *action-controllable* at every timestep, distinct from "real-time" (high FPS). For diffusion models, real-time FPS is often achieved during post-training by distilling the base model into fewer denoising steps [1]. While distillation is outside this paper's scope, PERSIST is fully compatible with these techniques.
> > >
> > > Furthermore, we demonstrate that PERSIST maintains good generation quality even with drastically reduced denoising steps and without any post-training, which we attribute to the strong generation stability provided by the 3D representation. Running PERSIST with just 2 denoising steps for the world-frame and 4 for the pixel-frame yields an inference speed of 886 ms (**1.13 FPS**). The resulting FVD is **256/266/274 at 200/400/600 frames**, outperforming the PixelDiT/WorldMem baselines employing 20 denoising steps. This demonstrates that PERSIST's 3D guidance allows it to operate with significantly fewer steps, effectively offsetting the computational overhead of tracking an explicit 3D state.
> > >
> > > * **PERSIST fast inference sample episode:** https://figshare.com/s/ffa6db3b5f561462a678
> > >
> > > **3. Insufficient Ablation Metrics**
> > > We hope the additional experiments provided above help address the reviewer's concerns by demonstrating the impact of our approach on both generation quality (evidenced by the FVD analysis on the 200-frame set) and temporal stability and consistency (evidenced by the extended FVD analysis, qualitative comparisons, and example rollouts). We hope the example rollouts provided give the reviewer an indication of the improved 3D consistency of our method. To supplement this, *we are fully committed to repeating our human study for all ablations in the camera-ready version to strenghten these results.*
> > >
> > > We hope these comprehensive additions resolve any remaining concerns, and we respectfully ask the reviewer to consider updating their score.
> > >
> > > [1] Yin, Tianwei et al. “From Slow Bidirectional to Fast Autoregressive Video Diffusion Models.” CVPR (2024).

---

### Official Review · Reviewer_4t61 · 2026-03-13

**Soundness:** 3
**Presentation:** 3
**Significance:** 2
**Originality:** 2
**Overall Recommendation:** 4
**Confidence:** 5

**Summary:**

The paper proposes an interactive world-modeling framework that replaces short pixel-history conditioning and key-frame retrieval with a persistent latent 3D state. The model decomposes simulation into three parts: predicting a latent world-frame, predicting the camera/viewpoint, and rendering the next frame from the 3D state through differentiable projection plus a learned screen-space renderer. The main claim is that explicitly modeling a latent 3D world improves long-horizon spatial memory and temporal consistency compared with standard autoregressive video models that only condition on past RGB frames.

**Compliance With Llm Reviewing Policy:**

Affirmed.

**Final Justification:**

Thanks the reviewers for their rebuttal efforts. I do not have much issue with the paper initially and will maintain my score.

**Key Questions For Authors:**

See weaknesses.

**Limitations:**

yes

**Strengths And Weaknesses:**

Strengths:
- Moving from pixel memories to a persistent latent 3D world state is a natural way to address long-horizon spatial memory and consistency in interactive generation, makes sense.
- The paper treats the camera as a query into the world state, then projecting 3D features into screen space before pixel generation, gives the method a strong geometric rationale rather than relying on heuristic RGB memory retrieval.
- The empirical gains are strong and consistent.
- The paper highlights interesting capabilities beyond better scores, especially explicit 3D initialization and continued off-screen dynamics.

Weaknesses:
- Not a weakness for this paper but a weakness for this series of work: this class of methods depend on fairly strong supervision assumptions during training: the authors assume access to ground-truth 3D world information and camera poses. That limits the current applicability beyond game engines, simulators, or datasets with explicit 3D annotations. I'd like to hear about what the authors think about this, is it solvable?
- The empirical scope is still somewhat narrow. The main baselines are Oasis and WorldMem in a voxel-world setting.
- The authors might consider adding much more papers to its related work discussion, especially both explicit and implicit long-context video gen/world model papers:
  - VMem: Consistent interactive video scene generation with surfel-indexed view memory
  - Context as Memory: Scene-consistent interactive long video generation with memory retrieval
  - Mixture of Contexts for long video generation
  - StateSpaceDiffuser: Bringing long context to diffusion world models
  - Long-context state-space video world models
  - One-minute video generation with test-time training; Test-time training done right
  - Frame context packing and drift prevention in next-frame-prediction video diffusion models
  - Pretraining frame preservation in autoregressive video memory compression
  - VideoSSM: Autoregressive long video generation with hybrid state-space memory

---

> ### Author Rebuttal · Authors · 2026-03-31
>
> We sincerely thank the reviewer for their careful evaluation of our work and for recognizing the novel capabilities our method introduces. We hope our new experiments and revisions address your remaining concerns. (References are omitted below due to space limit.)
>
> To better emphasize the novel capabilities unlocked by our explicit 3D representation, we have added two new figures:
> 1. **Off-screen dynamics:** We demonstrate two scenarios in this anonymized figure (https://figshare.com/s/f51892b14195f8bba13a). The top example shows a waterfall modeled within the 3D state, which eventually floods the area and visually submerges the player. The bottom example illustrates how unobserved areas evolve organically: a deep-sea cave gradually floods while the player swims in the ocean above.
> 2. **Mid-episode 3D-edits:** Our explicit 3D representation enables novel control over the generated experience. We demonstrate the ability to precisely edit the 3D world mid-generation to alter the agent's environment. As shown in this anonymized figure (https://figshare.com/s/11d38c4b544c91c03a00), pausing at any timestep allows us to manually insert trees, a hill, or entirely new biomes, which integrate seamlessly once generation resumes.
>
> **Reply to Weakness 1**: The reviewer touches on an important challenge for this class of approaches. While our current method relies on 3D annotations, we believe this requirement can be relaxed through two promising avenues:
> * **Utilizing 2D-to-3D foundation models:** Recent advances (VGGT, SAM 3D) demonstrate the ability to extract high-quality synthetic 3D annotations and camera poses directly from videos. Integrating these models as a preprocessing step could provide scalable, synthetic 3D annotations for unannotated video datasets.
> * **Learning 3D priors in the post-training phase:** Similar to how GameFactory successfully decoupled appearance and controllability by fine-tuning a general video generation model on a small, specialized Minecraft dataset, we hypothesize a general video model could acquire strong 3D priors by fine-tuning on a comparably small fraction of 3D-annotated simulator trajectories, leaving the bulk of the training data as unannotated video.
>
> **Reply to Weakness 2**: We acknowledge this and have conducted ablations to validate each component of PERSIST. Given the time constraints of the discussion period, we report the FVD for these ablations on a single evaluation set (Free-play), and indicate when an experiment is still in progress. The final manuscript will include a comprehensive evaluation across all sets.
> * **a) Pixel DiT:** Since the screen-space denoiser $\mathcal{P}_\theta$ employs the same architecture as the Oasis DiT, the baseline discussed in above reply to Weakness 1 also acts as an ablation (no 3D state modelling). FVD significantly **degrades from 148 (PERSIST-XL) to 875**.
> * **b) No voxel upscaling:** We do not upscale the world-frame to its native resolution (using the W-VAE decoder) when projecting it to screen-space. **FVD 148 -> 247**.
> * **c) No camera model:** Replaces the learned camera model with ground-truth camera trajectories. **FVD 148 -> 152**. While the FVD does not change significantly, we can observe the limitations of conditioning on camera ground truth trajectories in this particular example (https://figshare.com/s/9ba7beb6be8e3829efde). The agent repeatedly clips through its surroundings because the initial 3D state generated by PERSIST will not necessarily match the original environment layout. It interferes with generation and results in an accumulation of visual artifacts. In contrast, the learned camera model ensure camera states remain physically consistent with the generated geometry.
> * **d) No exposure bias mitigation:** We remove the random noise augmentation applied during training (see Section 5.3). It is possible that training will not have completed by the end of the discussion period, as this ablation requires retraining both the pixel and world frame denoisers from scratch.
>
> **Reply to Weakness 3**: We thank the reviewer for these suggestions. While we cannot modify the manuscript now, we will incorporate the following comparison into the final version to contextualize PERSIST:
> * **Explicit vs. Implicit 3D**: Unlike retrieval-based indexing (VMem [1], Context as Memory [2]), PERSIST simulates the **evolution** of a latent 3D world-frame. This allows off-screen processes to continue evolving even when not directly observed.
> * **Long-Context Architectures**: While SSMs and Mixture-of-Contexts [3,4,5,9] handle temporal dependencies implicitly, PERSIST's explicit 3D modeling provides inherent geometric consistency often missing in purely latent temporal models.
> * **Memory Paradigm**: Instead of optimizing standard video models (TTT [6], Context Packing [7,8]), PERSIST shifts memory to a persistent 3D latent space. This resolves viewpoint-dependency and partial observability without requiring per-sequence optimization.

---

> > ### Author Rebuttal · Reviewer_4t61 · 2026-04-03
> >
> > I don't have much problem with the submission and will keep my score.

---

### Decision · Program_Chairs · 2026-04-30

**Decision:**

Accept (regular)

**Comment:**

All reviewers agreed that the proposed use of a latent 3D state, as opposed to pixel context history, is both a conceptually and technically novel and interesting idea for representing history and ensuring consistent future world modeling.

Concerns were primarily about related work (addressed during the rebuttal), ablations (addressed during the rebuttal), reliance on heavy supervision signals (acknowledged, but unaddressable -- this is an issue for many classes of model within this family), generality (evaluation is on a limited set of domains, which are not fully "real world" -- partially addressed during the rebuttal).

While there is room for improvement (in evaluating in more domains, and attempting to reduce supervisory signals), this paper presents a conceptually interesting idea and some initial promise that this idea could be useful for long consistent video generation. I therefore recommend acceptance.